# EFFICIENT COVARIANCE ESTIMATION FOR SPARSIFIED FUNCTIONAL DATA

## ABSTRACT

To avoid prohibitive computation cost of sending entire data, we propose four sparsification schemes RANDOM-KNOTS, RANDOM-KNOTS-SPATIAL, B-SPLINE, BSPLINE-SPATIAL, and present corresponding nonparametric estimation of the covariance function. The covariance estimators are asymptotically equivalent to the sample covariance computed directly from the original data. And the estimated functional principal components effectively approximate the infeasible principal components under regularity conditions. The convergence rate reflects that leveraging spatial correlation and B-spline interpolation helps to reduce information loss. Data-driven selection method is further applied to determine the number of eigenfunctions in the model. Extensive numerical experiments are conducted to illustrate the theoretical results. [1]

## 1 INTRODUCTION

Dimension reduction has received increasing attention to avoid expensive and slow computation. Stich et al. (2018) investigated the convergence rate of Stochastic Gradient Descent after sparsification. Jhunjhunwala et al. (2021) focused on the mean function of a vector containing only a subset of the original vector. The goal of this paper is to estimate the covariance function of sparsified functional data, which is a set of sparsified vectors collected from a distributed system of nodes.

Functional data analysis (FDA) has become an important research area due to its wide applications. Classical FDA requires a large number of regularly spaced measurements per subject. The data takes the form $\{(x_{ij}, j/d), 1 \le i \le n, 1 \le j \le d\}$ in which $x_i(\cdot)$ is a latent smooth trajectory,

$$x_i(\cdot) = m(\cdot) + Z_i(\cdot). \tag{1}$$

The deterministic function $m(\cdot)$ denotes the common population mean, the random $Z_i(\cdot)$ are subject-specific small variation with $\mathbb{E}Z_i(\cdot) = 0$. Both $m(\cdot)$ and $Z_i(\cdot)$ are smooth functions of time $t = j/d$ which is rescaled to domain $\mathcal{D} = [0, 1]$. Trajectories $x_i(\cdot)$ are identically distributed realizations of the continuous stochastic process $\{x(t), t \in \mathcal{D}\}$, $\mathbb{E}\sup_{t \in \mathcal{D}} |x(t)|^2 < +\infty$ which can be decomposed as $x(\cdot) = m(\cdot) + Z(\cdot)$, $\mathbb{E}Z(t) = 0$. The true covariance function is $G(t, t') = \text{Cov}\{Z(t), Z(t')\}$.

Let sequences $\{\lambda_k\}_{k=1}^{\infty}$ and $\{\psi_k\}_{k=1}^{\infty}$ be the eigenvalues and eigenfunctions of $G(t, t')$, respectively, in which $\lambda_1 \ge \lambda_2 \ge \cdots \ge 0, \sum_{k=1}^{\infty} \lambda_k < \infty$, $\{\psi_k\}_{k=1}^{\infty}$ form an orthonormal basis of $L^2[0, 1]$, see Hsing & Eubank (2015). Mercer Lemma entails that the $\psi_k$'s are continuous and $G(t, t') = \sum_{k=1}^{\infty} \lambda_k \psi_k(t) \psi_k(t')$, $\int G(t, t') \psi_k(t') dt' = \lambda_k \psi_k(t)$. The standard process $x(\cdot)$ allows Karhunen-Loève $L^2$ representation $x(\cdot) = m(\cdot) + \sum_{k=1}^{\infty} \xi_k \phi_k(\cdot)$, in which the random coefficients, $\xi_k$, called functional principal component (FPC) scores, are uncorrelated with mean 0 and variance 1. The rescaled eigenfunctions, $\phi_k$, called FPC, satisfy $\phi_k = \sqrt{\lambda_k} \psi_k$ and $\int \{x(t) - m(t)\} \phi_k(t) dt = \lambda_k \xi_k$, for $k \ge 1$. Although the sequences $\{\lambda_k\}_{k=1}^{\infty}$, $\{\phi_k\}_{k=1}^{\infty}$ and $\{\xi_{ik}\}_{i=1, k=1}^{n, \infty}$ exist mathematically, they are either unknown or unobservable.

### 1.1 MAIN CONTRIBUTION

In FDA, covariance estimation plays a critical role in FPC analysis (Ramsay & Silverman (2005), Li & Hsing (2010)), functional generalized linear models and other nonlinear models (Yao et al.

---

[1] The code is attached to the supplementary material and will be publicly available once accepted.

(2005b)). We propose four sparsification schemes, RANDOM-KNOTS & RANDOM-KNOTS-SPATIAL can be classified as RANDOM-SPARSIFICATION where knots are selected uniformly from the entire points at random. B-SPLINE & BSPLINE-SPATIAL are called FIXED-SPARSIFICATION, intercepting knots at fixed positions in each dimension of the vector. For all sparsification modes, we construct the two-step covariance estimator $\hat{G}(\cdot,\cdot)$, where the first step involving sparsified trajectories and the second step plug-in covariance estimator by using the estimated trajectories in place of the latent true trajectories. The statistic is further multiplied by an appropriate constant to ensure unbiasedness. The covariance estimator $\hat{G}(\cdot,\cdot)$ enjoys good properties and can effectively approximates the sample covariance function $\bar{G}(\cdot,\cdot)$ computed directly from the original data. This paper improves the performance of statistics from the following two aspects, requiring little or no side information and additional local computation.

- SPATIAL CORRELATION: We adjust the fixed weights assigned to different vectors to data-driven parameters that represent the amount of spatial correlation. This family of statistics naturally takes the influence of correlation among nodes into account, which can be viewed as spatial factor [2]. Theoretical derivation reveals that the estimation error can be drastically reduced when spatial factors among subjects are considered.
- B-SPLINE INTERPOLATION: To fill in the gap between equispaced knots, we introduce spline interpolation method to characterize the temporally ordered trajectories of the functional data. The estimated trajectory obtained by B-spline smoothing method is as efficient as the original trajectory. The proposed covariance estimator has globally consistent convergence rate, enjoying superior theoretical properties than that without interpolation. Superior to the covariance estimation leveraging tensor product B-splines, our two-step estimators are guaranteed to be the positive semi-definite.

In sum, the main advantage of our methods is the computational efficiency and feasibility for large-scale dense functional data. It is practically relevant since curves or images measured using new technology are usually of much higher resolution than the previous generation. This directly leads to the doubling of the amount of data recorded in each node, which is also the motivation of this paper to propose sparsification before feature extraction, modeling, or other downstream steps.

The paper is organized as follows. Section 2 introduces four sparsification schemes and the corresponding unbiased covariance estimators. We also deduce the convergence rate of the covariance estimators and the related FPCs. Simulation studies are presented in Section 3 and application in domain clustering is in Section 4. All technical proofs are involved in the Appendix.

## 1.2 RELATED WORK

Considerable efforts have been made to analyze first-order structure of function-valued random elements, i.e., the functional mean $m(\cdot)$. Estimation of mean function has been investigated in Jhunjhunwala et al. (2021), Garg et al. (2014), Suresh et al. (2017), Mayekar et al. (2021) and Brown et al. (2021). Cao et al. (2012) and Huang et al. (2022) considered empirical mean estimation using B-spline smoothing. The second-order structure of random functions – covariance function $G(\cdot,\cdot)$ is the next object of interests. To the best of our knowledge, spatial correlation across nodes has not yet been considered in the context of sparsified covariance estimation. The research on sparsification has received wide attention recently, for instance Alistarh et al. (2018), Stich et al. (2018) and Sahu et al. (2021). Sparsification methods mainly focus on sending only a subset of elements of the vectors, yet no existing method combine sparsity method with B-spline fitting. Moreover, there has been striking improvement over sparse PCA. Berthet & Rigollet (2013b) and Choo & d'Orsi (2021) have analyzed the complexity of sparse PCA; Berthet & Rigollet (2013a) and Deshpande & Montanari (2014) have obtained sparse principle components for particular data models. Since our estimation methods are innovative, the related study of PCA is proposed for the first time.

## 2 MAIN RESULTS

We consider $n$ geographically distributed nodes, each node generates a $d$-dimensional vector $x_i = (x_{i1}, \ldots, x_{id})^\top$ for $i \in \{1, 2, \ldots, n\}$. The mean and covariance functions could be estimated

---

[2]The superscript SPAT is applied to distinguish whether the statistic considering spatial factor.

by $\bar{m}(t) = \frac{1}{n} \sum_{i=1}^{n} x_i(t)$ and $\bar{G}(t, t') = \frac{1}{n} \sum_{i=1}^{n} (x_i(t) - \bar{m}(t))(x_i(t') - \bar{m}(t'))$, see Brockwell & Davis (2009). For $q \in \mathbb{N}$, $\mu \in (0, 1]$, write $\mathcal{H}^{(q,\mu)}[0, 1]$ as the space of $\mu$-Hölder continuous functions, i.e., $\mathcal{H}^{(q,\mu)}[0, 1] = \left\{ \varphi : [0, 1] \to \mathbb{R} \mid \|\varphi\|_{q,\mu} = \sup_{x,y \in [0,1], x \neq y} \frac{|\varphi^{(q)}(x) - \varphi^{(q)}(y)|}{|x - y|^{\mu}} < +\infty \right\}$.

**Notations:** In this paper, $\mathcal{O}_p$ (or $o_p$) denotes a sequence of random variables of certain order in probability and by $\mathcal{O}_{a.s.}$ (or $o_{a.s.}$) almost surely $\mathcal{O}$ (or $o$). For sequences $a_n$ and $b_n$, denote $a_n \asymp b_n$ if $a_n$ and $b_n$ are asymptotically equivalent. For any Lebesgue measurable function $\phi(\mathbf{x})$ on a domain $\mathcal{D}$, let $\|\phi\|_{\infty} = \sup_{\mathbf{x} \in \mathcal{D}} |\phi(\mathbf{x})|$. For any $L^2$ integrable functions $\phi(\mathbf{x})$ and $\varphi(\mathbf{x})$, $\mathbf{x} \in \mathcal{D}$, take $\langle \phi, \varphi \rangle = \int_{\mathcal{D}} \phi(\mathbf{x})\varphi(\mathbf{x})d\mathbf{x}$, with $\|\phi\|_2^2 = \langle \phi, \phi \rangle$. For simplicity, $\|\phi\| = \|\phi\|_2$.

We next introduce some technical assumptions.

**Assumption 1:** There exists an integer $q > 0$ and a constant $\mu \in (0, 1]$, such that the regression function $m(\cdot) \in \mathcal{H}^{(q,\mu)}[0, 1]$. In the following, we denote $p^* = q + \mu$ for simplicity.

**Assumption 2:** The covariance function satisfies $\sup_{(t,t') \in [0,1]^2} G(t, t') < C$, for some positive constant $C$ and $\min_{t \in [0,1]} G(t, t') > 0$.

**Assumption 3:** There exists a constant $\theta > 0$, such that as $d \to \infty$, $n = n(d) \to \infty$, $n = \mathcal{O}(d^{\theta})$.

**Assumption 4:** The rescaled FPCs $\phi_k(\cdot) \in \mathcal{H}^{(q,\mu)}[0, 1]$ with $\sum_{k=1}^{\infty} \left( \|\phi_k\|_{q,\mu} + \|\phi_k\|_{\infty} \right) < +\infty$; for increasing positive integers $\{k_n\}_{n=1}^{\infty}$, as $n \to \infty$, $\sum_{k_n+1}^{\infty} \|\phi_k\|_{\infty} = \mathcal{O}(n^{-1/2})$ and $k_n = \mathcal{O}(n^{\omega})$ for some $\omega > 0$.

**Assumption 5:** The FPC scores $\{\xi_{ik}\}_{i \geq 1, k \geq 1}$ are independent over $k \geq 1$. The number of distinct distributions for all FPC scores $\{\xi_{ik}\}_{i \geq 1, k \geq 1}$ is finite, and $\max_{1 \leq k < \infty} \mathbb{E}\xi_{1k}^{r_0} < \infty$ for $r_0 > 4$.

**Assumption 6:** The number of knots $J_s \asymp d^{\gamma} C_d$ for some $\tau > 0$ with $C_d + C_d^{-1} = \mathcal{O}(\log^{\tau} d)$ as $d \to \infty$, $\gamma \geq 1 - \frac{\theta}{2}$ for RANDOM-SPARSIFICATION, $\gamma > \frac{\theta}{2p^*} + \frac{2\theta}{r_0 p^*}$ for FIXED-SPARSIFICATION.

Assumptions 1–5 are standard requirements for obtaining mean and covariance estimators in literature. Assumption 1 guarantees the orders of the bias terms of the spline smoothers for $m(\cdot)$. Assumption 2 ensures the covariance $G(\cdot, \cdot)$ is uniformly bounded. Assumption 3 implies the dimension of vectors $d$ diverges to infinity as $n \to \infty$, which is a well-developed asymptotic scenario for dense functional data and all the following asymptotics are developed as both the number of nodes $n$ and dimensionality $d$ tend to infinity. Assumption 4 concerns the bounded smoothness of FPC and Assumption 5 ensures bounded FPC scores, for bounding the bias terms in the spline covariance estimator. The smoothness of our estimator is controlled by the number of knots, which is specified in Assumption 6.

**Remark 1.** *These assumptions are mild conditions that can be satisfied in many practical situations. One simple and reasonable setup is: $q + \mu = p^* = 4$, $\theta = 1$, $\gamma = 5/8$, $C_d \asymp \log \log d$. We set $p = 4$ for RANDOM-SPARSIFICATION to obtain cubic spline estimation and $p = 0$ for FIXED-SPARSIFICATION to get estimates without interpolation. These constants are used as defaults in numerical studies.*

## 2.1 RANDOM-SPARSIFICATION

Elements in each vector are randomly setting to zero with probability $1 - \frac{J_s}{d}$. Under this sparsification scheme, RANDOM-KNOTS and RANDOM-KNOTS-SPATIAL covariance estimators are proposed, where temporal dependence exhibited in functional samples is ignored. Proportion $\frac{J_s}{d}$ depicts the difference of data volume before and after sparsification, reflecting the degree of sparsification.

RANDOM-KNOTS (RK) As shown in Figure 2.2 (a), on the left are original vectors $\{x_i\}_{i=1}^{n}$ and right are sparsified vectors $\{h_i\}_{i=1}^{n}$ randomly containing $J_s$ elements of the original vector with $n = 3$, $d = 6$, $J_s = 3$. This definition tells that $\mathbb{P}(h_{ij} = 0) = 1 - \frac{J_s}{d}$, $\mathbb{P}(h_{ij} = x_{ij}) = \frac{J_s}{d}$. The estimator generated from $\{h_i\}_{i=1}^{n}$ is called sparsified estimator which is obtained by replacing original trajectory $x_i = (x_{i1}, \ldots, x_{id})^{\top}$ with sparsified $h_i = (h_{i1}, \ldots, h_{id})^{\top}$, i.e. $\hat{m} = \frac{1}{n} \frac{d}{J_s} \sum_{i=1}^{n} h_i$ and $\hat{G} = \frac{1}{n} \left( \frac{d}{J_s} \right)^2 \sum_{i=1}^{n} (h_i - \bar{h})(h_i - \bar{h})^{\top}$, where $\bar{h} = \frac{1}{n} \sum_{i=1}^{n} h_i$ is a $d$-dimensional vector.

The next theorem states the mean squared error (MSE) of the estimator tends to zero as $n \to \infty$.

**Theorem 1.** *(RK Estimation Error) Under Assumptions 1–5, MSE of estimate $\hat{G}$ produced by the RK sparsification scheme described above is given by $\mathbb{E}\|\hat{G} - \bar{G}\|^2 = \frac{1}{n^2}\left(\left(\frac{d}{J_s}\right)^2 - 1\right)R_1$ where $R_1 = \sum_{i=1}^n \|x_i - \bar{m}\|^4$. Assumption 6 further guarantees that $\left\|\hat{G} - \bar{G}\right\| = \mathcal{O}_p\left(n^{-1/2}\right)$.*

RANDOM-KNOTS-SPATIAL (RK-SPAT) Denote by $M_j$ the number of nodes that send their $j$-th coordinate to describe the correlation between nodes. It is obvious that $M_j$ is a binomial random variable that takes values in the range $\{0, 1, \ldots, n\}$ with $\mathbb{P}\left(M_j = m\right) = \binom{n}{m}p^m\left(1 - p\right)^{n-m}$, $p = \frac{J_s}{d}$. If $M_j = 0$, none of the nodes have drawn the $j$-th element, and the information at position $j$ is completely missing.

If nodes are highly correlated, the estimator is accurate at position $j$ even if few points at that position are selected. Consider a special case where vectors of all nodes are the same, i.e., $x_1 = x_2 = \ldots = x_n$. The $j$-th coordinate of mean can be exactly estimated as $\hat{m}_j = \frac{1}{M_j}\sum_{i=1}^n h_{ij}$ whenever $M_j > 0$. And the exact covariance function is $\hat{G}_{jj'} = \frac{1}{M_j}\sum_{i=1}^n\left(h_{ij} - \bar{h}_j\right)\left(h_{ij'} - \bar{h}_{j'}\right)$, $\bar{h}_j$ is the $j$-th element of the $d$-dimensional vector $\bar{h}$, $j' \neq j$. Simple mathematical derivation implies that $\bar{h}_j = \hat{m}_j$ under this situation. Hence, the fixed scaling parameter $\frac{J_s}{d}$ is not necessary.

If nodes are lowly correlated, small $M_j$ may lead to a large MSE. Consider (i) vectors corresponding to $n - 1$ nodes follow sine distribution $x_{ij} = \sin\left(2\pi\frac{j}{d}\right)$, $1 \leq i \leq n - 1$, $1 \leq j \leq d$ and the outlier node follows cosine distribution $x_{nj} = \cos\left(2\pi\frac{j}{d}\right)$; (ii) the outlier vector has a jump at $j$-th position $x_{nj} = \sin\left(2\pi\frac{j}{d}\right) + \delta$, $x_{nj'} = \sin\left(2\pi\frac{i'}{d}\right)$, $\delta > 0$, for $j' \in \{1, \ldots j-1, j+1, \ldots, d\}$ while other $n - 1$ nodes follow standard sine distribution. In special case that only the outlier vector is selected for position $j$, the estimation at this position is bound to have large deviation. Therefore, the correlation between nodes is an important indicator to measure the accuracy of estimators.

We propose a RK-SPAT estimator wherein the fixed scaling parameter $\frac{J_s}{d}$ is replaced by a function of $M_j$ such that the spatial correlation between nodes is taken into account. Specifically, the mean estimator for $j$-th element is $\hat{m}_j^{\text{SPAT}} = \frac{1}{n}\frac{\bar{\beta}}{T(M_j)}\sum_{i=1}^n h_{ij}$. Covariance function at position $(j, j')$ is

$$\hat{G}_{jj'}^{\text{SPAT}} = \frac{1}{n}\frac{\bar{\beta}^2}{T(M_j)T(M_{j'})}\sum_{i=1}^n\left(h_{ij} - \bar{h}_j\right)\left(h_{ij'} - \bar{h}_{j'}\right) \tag{2}$$

where the introduced function $T(M_j)$ changes the scaling parameter and $\bar{\beta}$ is

$$\bar{\beta}^{-1} = \frac{J_s}{d}\mathbb{E}_{M_j|M_j \geq 1}\left(\frac{1}{T(M_j)}\right) = \sum_{r=1}^n \frac{J_s}{dT(r)}\binom{n-1}{r-1}\left(\frac{J_s}{d}\right)^{r-1}\left(1 - \frac{J_s}{d}\right)^{n-r}. \tag{3}$$

We prove that the RK-SPAT covariance estimator is unbiased.

**Proposition 1.** *(RK-SPAT estimator Unbiasedness) $\mathbb{E}\hat{G}^{\text{SPAT}} = \bar{G}$.*

The following theorem measures the approximate quality of RK-SPAT estimator,

**Theorem 2.** *(RK-SPAT Estimation Error) Under Assumptions 1–5, MSE of estimate produced by the RK-SPAT family is $\mathbb{E}\|\hat{G}^{\text{SPAT}} - \bar{G}\|^2 = \frac{1}{n^2}\left(\left(\frac{d}{J_s} + c_1\right)^2 - 1\right)R_1 + \frac{1}{n^2}\left(\left(1 - c_2\right)^2 - 1\right)R_2$, where $R_1 = \sum_{i=1}^n \|x_i - \bar{m}\|^4$, $R_2 = 2\sum_{i=1}^n\sum_{k=i+1}^n\left\langle\left(x_i - \bar{m}\right)^2, \left(x_k - \bar{m}\right)^2\right\rangle$ and $\bar{\beta}$ is defined in (3). The parameters $c_1$, $c_2$ depend on the choice of $T(\cdot)$ as*

$$c_1 = \bar{\beta}^2\sum_{r=1}^n \frac{J_s}{dT(r)^2}\binom{n-1}{r-1}\left(\frac{J_s}{d}\right)^{r-1}\left(1 - \frac{J_s}{d}\right)^{n-r} - \frac{d}{J_s}$$

$$c_2 = 1 - \bar{\beta}^2\sum_{r=2}^n \frac{J_s^2}{d^2 T(r)^2}\binom{n-2}{r-2}\left(\frac{J_s}{d}\right)^{r-2}\left(1 - \frac{J_s}{d}\right)^{n-r}.$$

*Assumption 6 further guarantees that $\left\|\hat{G}^{\text{SPAT}} - \bar{G}\right\| = \mathcal{O}_p\left(n^{-1/2}\right)$.*

Theorem 2 can be further simplified as $\mathbb{E}\|\hat{G}^{\text{SPAT}} - \bar{G}\|^2 = \frac{1}{n^2}\left(\left(\frac{d}{J_s}\right)^2 + c_1^2 + 2c_1\frac{d}{J_s} - 1\right)R_1 + \frac{1}{n^2}\left(c_2^2 - 2c_2\right)R_2$. The MSE of RK-SPAT covariance estimator ensures to be lower than that of RK estimator whenever $\left(c_1^2 + 2c_1\frac{d}{J_s}\right)R_1 < \left(2c_2 - c_2^2\right)R_2$, i.e. $R_2/R_1 > \left(c_1^2 + 2c_1\frac{d}{J_s}\right)/\left(2c_2 - c_2^2\right)$. In general, since the MSE depends on the function $T(\cdot)$ through $c_1$ and $c_2$, we can define $T(\cdot)$ to ensure that RK-SPAT estimate is more accurate than RK estimate.

**Theorem 3.** *(RK-SPAT minimum MSE) The optimal RK-SPAT estimator that minimizes the MSE in Theorem 2, can be obtained by setting* $T^*(r) = \left(1 + \frac{R_2}{R_1}\left(\frac{r-1}{n-1}\right)^2\right)^{1/2}$, *for* $r \in \{1, 2, \ldots, n\}$.

Jhunjhunwala et al. (2021) claimed that optimal RK-SPAT mean estimator is obtained when $T^*(r) = 1 + \frac{R_2}{R_1}\frac{r-1}{n-1}$, which is perfect for tasks involving only mean function, such as K-means but cannot guarantee the optimal estimation of covariance function. It is significant to get accurate covariance estimate by setting $T^*(\cdot)$ as Theorem 3. In this way, the extracted features computed from eigenequation would improve the efficiency of downstream tasks, such as PCA. Meanwhile, the nodes number $n$ and dimension $d$ yield the amount of computation for $R_1$ and $R_2$. We propose the RK-SPAT (AVG) $\tilde{T}(r) = \left(1 + \frac{n}{2}\left(\frac{r-1}{n-1}\right)^2\right)^{1/2}$ as a default setting to avoid complicated computation about $R_1$ and $R_2$.

## 2.2 FIXED-SPARSIFICATION

We retain elements at $J_s$ fixed positions and set the rest to zero. This dimensionality reduction method only utilizes values at fixed positions in the vector and has several disadvantages. (i) Each step only leverages the subset of data with size $n \times J_s$, while the size of the origin data set is $n \times d$. The fact that $J_s \ll d$ resulting in a serious loss of information. (ii) Approximate quality of the estimator depends on the selected points, and is difficult to control if the selected knots deviate from the overall distribution. (iii) Even if suitable knots are determined by adding penalty terms or selecting artificially, these knots may not be suitable for another node.

B-SPLINE (BS) As shown in Figure 2.2 (b), B-spline interpolation reduces the loss of information by fitting the values between fixed knots. It is worth noticing that the choice of basis functions and other smoothing methods (polynomial, kernel and wavelet smoothing) do not affect the large-sample theories. We choose standard B-spline bases because they are more computationally efficient and numerically stable in finite samples compared with other basis functions such as the truncated power series and trigonometric series. The B-spline estimation is suitable for analyzing large data sets without uniform distribution, see Schumaker (2007).

Denote by $\{t_\ell\}_{\ell=1}^{J_s}$ a sequence of equally-spaced points, $t_\ell = \ell/(J_s + 1)$, $0 < t_1 < \cdots < t_{J_s} < 1$, called interior knots, which divide the interval $[0,1]$ into $(J_s + 1)$ equal subintervals $I_0 = [0, t_1)$, $I_\ell = [t_\ell, t_{\ell+1})$, $\ell \in \{1, \ldots, J_s - 1\}$, $I_{J_s} = [t_{J_s}, 1]$. Let $t_{1-p} = \cdots = t_0 = 0$, $1 = t_{J_s+1} = \cdots = t_{J_s+p}$ be auxiliary knots, and $\mathcal{S}^{(p-2)} = \mathcal{S}^{(p-2)}[0,1]$ be the polynomial spline space of order $p$ on $I_\ell$, $\ell \in \{0, \ldots, J_s\}$, which consists of all $(p-2)$ times continuously differentiable functions on $[0,1]$ that are polynomials of degree $(p-1)$ on subintervals $I_\ell$. Denote by $\{B_{\ell,p}(t), 1 \leq \ell \leq J_s + p\}$ the $p$-th order B-spline basis functions of $\mathcal{S}^{(p-2)}$, hence $\mathcal{S}^{(p-2)} = \left\{\sum_{\ell=1}^{J_s+p}\lambda_{\ell,p}B_{\ell,p}(t) \mid \lambda_{\ell,p} \in \mathbb{R}, t \in [0,1]\right\}$. The unknown trajectory $x_i(\cdot)$ is estimated by $h_i(\cdot) = \text{argmin}_{g(\cdot) \in \mathcal{S}^{(p-2)}}\sum_{j=1}^d\left\{x_{ij} - g(x_j)\right\}^2 = \sum_{\ell=1}^{J_s+p}\hat{\lambda}_{\ell,p,i}B_{\ell,p}(\cdot)$. Coefficients estimation follows $\left(\hat{\lambda}_{1,p,i}, \ldots, \hat{\lambda}_{J_s+p,p,i}\right)^\top = \text{argmin}_{(\lambda_{1,p}, \ldots, \lambda_{J_s+p,p}) \in \mathbb{R}^{J_s+p}}\sum_{j=1}^d\left\{x_{ij} - \sum_{\ell=1}^{J_s+p}\lambda_{\ell,p}B_{\ell,p}(j/d)\right\}^2$. The BS covariance estimator is obtained by replacing $\{h_{ij}\}_{i=1,j=1}^{n,d}$ with B-spline trajectories.

BSPLINE-SPATIAL (BS-SPAT) We replace $\{h_{ij}\}_{i=1,j=1}^{n,d}$ in RK-SPAT estimator with B-spline estimation of trajectories. The BS-SPAT estimator not only considers the correlation among nodes, but also the correlation within a single node.

Next theorem states the convergence rate of BS and BS-SPAT estimators.

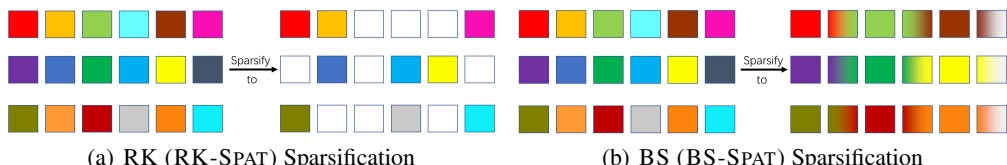

(a) RK (RK-Spat) Sparsification         (b) BS (BS-Spat) Sparsification

**Theorem 4.** *(BS (BS-Spat) Estimation Error) Under Assumptions 1–6, the BS (BS-Spat) estimator $\hat{m}(\cdot)$ is asymptotically equivalent to $\bar{m}(\cdot)$ up to order $n^{1/2}$ and similar conclusion holds for covariance function, i.e., as $n \to \infty$,*

*(i) $\|\hat{m} - \bar{m}\|_\infty = \mathcal{O}_{a.s.}\left(n^{-1/2}\right)$, $\|\hat{G} - \bar{G}\|_\infty = \mathcal{O}_p\left(n^{-1/2}\right)$.*

*(ii) $\|\hat{m}^{\text{Spat}} - \bar{m}\|_\infty = \mathcal{O}_p\left(n^{-1/2}\right)$, $\|\hat{G}^{\text{Spat}} - \bar{G}\|_\infty = \mathcal{O}_p\left(n^{-1/2}\right)$.*

**Remark 2.** *The convergence rates in Theorems 1, 2 and 4 reflect that BS (BS-Spat) covariance estimator converges faster to $\bar{G}$ than RK (RK-Spat) estimator and the result uniformly holds on the interval $\mathcal{D}$. It is obvious that the estimation performance of the proposed estimator is significantly improved by applying B-spline interpolation. This conclusion is also confirmed in the simulation.*

### 2.3 Convergence of Principal Component

The estimates of eigenfunctions and eigenvalues are obtained by solving the eigenequations $\int \hat{G}(x, x')\hat{\psi}_k(x')\,dx' = \hat{\lambda}_k \hat{\psi}_k(x)$, the consistency of which is then obtained.

**Theorem 5.** *As $n \to \infty$, for $k \in \mathbb{N}$, we have*

*(i) (Convergence rate of eigenfunctions) $\left\|\hat{\psi}_k - \psi_k\right\| = \mathcal{O}_p\left(n^{-1/2}\right)$;*

*(ii) (Convergence rate of eigenvalues) $\left|\hat{\lambda}_k - \lambda_k\right| = \mathcal{O}_p\left(n^{-1/2}\right)$;*

*(iii) (Convergence rate of FPC scores) $\max_{1 \leq i \leq n}\left\|\hat{\xi}_{ik} - \xi_{ik}\right\| = \mathcal{O}_p\left(n^{-1/2}\right)$.*

It is worth noticing that the orthonormal basis of the eigenmanifold corresponding to $\{\lambda_k\}_{k=1}^\kappa$ may be obtained by rotation, see Dauxois et al. (1982). Therefore, the unique form of the eigenfunction should be determined by minimizing the estimation error through the loss function $L(\hat{\phi}_k, \phi_k) = \frac{1}{2}\min_{s\in\{+1,-1\}}\|\hat{\phi}_k - s\phi_k\|^2 = 1 - |\langle\hat{\phi}_k, \phi_k\rangle|$ for $\hat{\phi}_k, \phi_k \in \{\mathbf{v} \in \mathbb{R}^\kappa : \|\mathbf{v}\| = 1\}$. This is required because the estimated principal components $\left\{\hat{\phi}_k\right\}_{k=1}^\kappa$ are only identifiable up to a sign. Analogous results can obtained for alternate loss functions such as the projection distance: $L_p(\hat{\phi}_k, \phi_k) = \frac{1}{\sqrt{2}}\left\|\hat{\phi}_k\hat{\phi}_k^\top - \phi_k\phi_k^\top\right\|_2 = \sqrt{1 - \langle\hat{\phi}_k, \phi_k\rangle^2}$.

## 3 Simulation

We conduct simulation studies to illustrate the finite-sample performance of the proposed methods.

### 3.1 Knots selection

The number of knots is treated as an unknown tuning parameters, and the fitting results can be sensitive to it. Since the in-sample fitting errors cannot gauge the prediction accuracy of the fitted function, we select a criterion function that attempts to measure the out-of-sample performance of the fitted model. The formula based selection strategy as stated in Remark 1, we seek $J_s$ satisfies $J_s \asymp d^\gamma C_d$. Therefore, we suggest $J_s = [cd^\gamma \log\log d]$ for some positive constant $c$, and $\gamma = 5/8$.

Apart from this, minimizing Akaike information criterion (AIC) is one computationally efficient approach to selecting smoothing parameters. The candidate pool is all the integers between 1 and $J_{s^*}$, where $J_s^* = \min\{10, [cd^\gamma \log\log d]\}$. Specifically, given any data set $(x_{ij}, j/d)_{i=1,j=1}^{n,d}$ from

model (1), denote the estimator for response $x_{ij}$ by $h_{ij}(J)$. The trajectory estimates depend on the knot selection sequence, which are sparsified vectors for RK (RK-SPAT) estimator and B-spline smoothing vectors for BS (BS-SPAT) estimator. Then, $\hat{J}_{s,i}$ for the $i$-th curve is the one minimizing AIC,

$$\hat{J}_{s,i} = \operatorname{argmin}_{J \in [1, J_{s*}]} \operatorname{AIC}(J),$$

where $\operatorname{AIC}(J) = \log(\operatorname{RSS}/d) + 2(J+p)/d$, with the residual sum of squares $\operatorname{RSS} = \sum_{j=1}^{d} \{x_{ij} - h_{ij}(J)\}^2$. Then, $\hat{J}_s$ is set as the median of $\{\hat{J}_{s,i}\}_{i=1}^{n}$.

## 3.2 ACCURACY OF COVARIANCE ESTIMATOR

Data is generated from model:

$$x_{ij} = m(j/d) + \sum_{k=1}^{\infty} \xi_{ik} \phi_k(j/d), \quad 1 \le j \le d, \quad 1 \le i \le n, \quad k \ge 1, \tag{4}$$

where $m(t) = \sin\{2\pi(t-1/2)\}$, $\phi_k(t) = \sqrt{\lambda_k}\psi_k(t)$, $\lambda_k = (1/4)^{[k/2]}$, $\psi_{2k-1}(t) = \sqrt{2}\cos(2k\pi t)$, $\psi_{2k}(t) = \sqrt{2}\sin(2k\pi t)$. $\{\xi_{ik}\}$ follow standardized normal distribution. The infinite series $G(t,t') = \sum_{k=1}^{\infty} \phi_k(t)\phi_k(t')$ is well approximated by finite sum $G(t,t') = \sum_{k=1}^{1000} \phi_k(t)\phi_k(t')$, according to the fraction of variance explained (FVE) criteria, $\operatorname{FVE} = \sum_{k=1}^{1000} \lambda_k / \sum_{k=1}^{\infty} \lambda_k > 1 - 10^{-10}$, see Yao et al. (2005a). $d$ (or $n$) is set to vary equally between 50 and 400 with fixed $n = 200$ (or $d = 200$). Each simulation is repeated 1000 times.

Denote by $\hat{G}_{j,j'}^s$ ($\bar{G}_{j,j'}^s$) the $s$-th replication of covariance $\hat{G}$ ($\bar{G}$) at position $(j,j')$ and $G$ the true covariance function. The average mean squared error (AMSE) is computed to assess the performance of the covariance estimators $\hat{G}(\cdot,\cdot)$ and $\bar{G}(\cdot,\cdot)$, which is defined as $\operatorname{AMSE}(\hat{G}) = \frac{1}{1000d^2} \sum_{s=1}^{1000} \sum_{j,j'=1}^{d} \left(\hat{G}_{j,j'}^s - \bar{G}_{j,j'}^s\right)^2$, $\operatorname{AMSE}(\bar{G}) = \frac{1}{1000d^2} \sum_{s=1}^{1000} \sum_{j,j'=1}^{d} \left(\bar{G}_{j,j'}^s - G_{j,j'}\right)^2$.

Figure 1 [3] shows that $\operatorname{AMSE}(\hat{G})$ decreases as $n$ increases, consistent with Theorems 1 and 2. $\operatorname{AMSE}(\hat{G})$ reveals a slow downward trend with the increase of $d$, mainly because the number of knots changes with $d$, affecting the performance of the covariance estimator. By setting $T(\cdot)$ as Theorem 3, AMSE of estimator that takes spatial factor into account is generally lower than estimator that does not. AMSE of BS (BS-SPAT) covariance estimator confirms Theorem 4 and estimation accuracy of the estimator is significantly improved by spline interpolation. Results on $\operatorname{AMSE}(\bar{G})$ are consistent with the fact that $\bar{G}$ converges to $G$ at the rate of $\mathcal{O}_p(n^{-1/2})$. Visualization of covariance functions is in Figure H in Appendix.

## 3.3 ACCURACY OF PRINCIPLE COMPONENTS

Spectral decomposition is truncated at $\kappa = 5$ according to the standard criteria called "pseudo-AIC", see Mu et al. (2008). The selected eigenvalues can explain over 95% of the total variation. That is, $\kappa = \operatorname{argmin}_{1-p \le \ell \le J_s} \left\{ \sum_{k=1-p}^{\ell} \hat{\lambda}_k / \sum_{k=1-p}^{J_s} \hat{\lambda}_k > 0.95 \right\}$ where $\left\{\hat{\lambda}_k\right\}_{k=1-p}^{J_s}$ are all nonnegative eigenvalues estimated in FPC analysis. Figure 3 illustrates the first five eigenfunctions which account for 68.2%, 17.0%, 4.3%, 4.3%, 4.0% of the total variation. The first figure shows a large difference overall the curve, depicting the trend of the covariance. The other four graphs have great fluctuations, and the frequency of fluctuations increases with $k$. The gap between the estimated FPC computed from the covariance estimators and the true FPC also increases with $k$.

Denote by $\hat{\lambda}_k^s$, $\hat{\phi}_k^s$ the $s$-th replication of $\hat{\lambda}_k$, $\hat{\phi}_k$, AMSEs of eigenvalues $\hat{\lambda}_k$ 's and the eigenfunctions $\hat{\phi}_k$ 's are defined as $\operatorname{AMSE}(\hat{\lambda}) = \frac{1}{1000\kappa} \sum_{s=1}^{1000} \sum_{k=1}^{\kappa} \left(\hat{\lambda}_k^s - \lambda_k\right)^2$, $\operatorname{AMSE}(\hat{\phi}) = \frac{1}{1000d\kappa} \sum_{s=1}^{1000} \sum_{j=1}^{d} \sum_{k=1}^{\kappa} \left\{\left(\hat{\phi}_k^s - \phi_k\right)(j/d)\right\}^2$.

---

[3]"d: Random-knots" $\operatorname{AMSE}(\hat{G})$ changes with $d$ and $\hat{G}$ is the RK estimator; "n: Random-knots-Spatial" $\operatorname{AMSE}(\hat{G})$ changes with $n$ and $\hat{G}$ is the RK-Spat estimator. Other curves are defined similarly.

Figure 2 reveals that $\text{AMSE}(\hat{\lambda})$ decreases with the increase of $n$, while the change with $d$ is small, $\text{AMSE}(\hat{\phi})$ exhibits the same regularity as $\text{AMSE}(\hat{\lambda})$, in accordance with Theorem 5. Whether or not to consider spatial factors has greater impact on the accuracy of eigenvectors than eigenvalues.

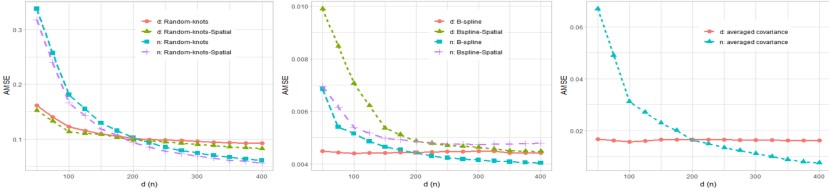

Figure 1: Left, middle: $\text{AMSE}(\hat{G})$ as a function of $d, n$. Right: $\text{AMSE}(\bar{G})$ as a function of $d, n$.

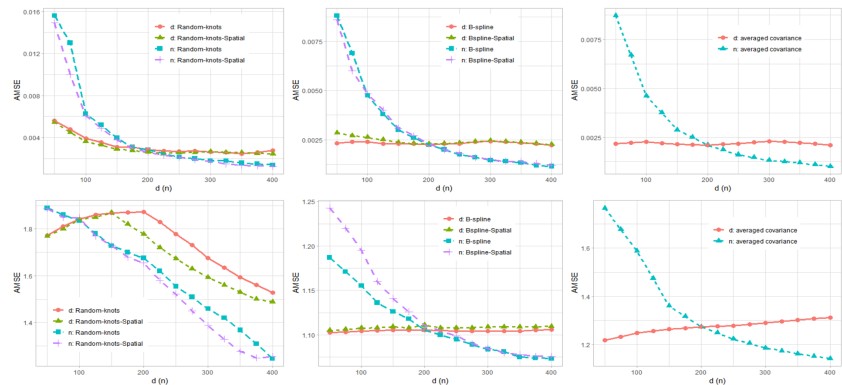

Figure 2: Row 1: $\text{AMSE}(\hat{\lambda})$ as a function of $d, n$. Row 2: $\text{AMSE}(\hat{\phi})$ as a function of $d, n$.

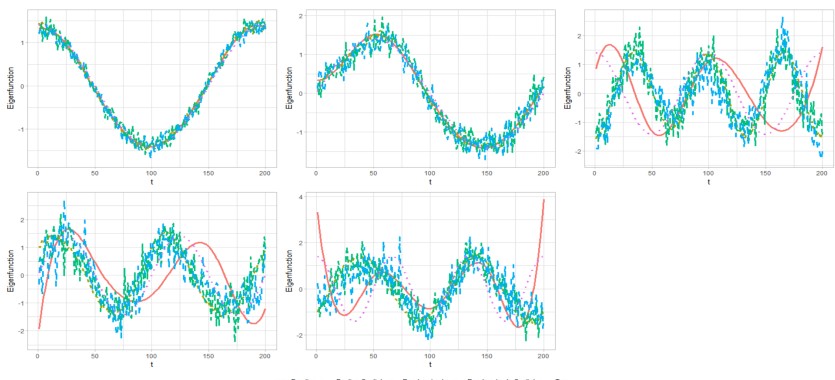

Figure 3: Eigenfunctions computed from the true covariance and four covariance estimators.

## 4 APPLICATION

We seek to harness our method for a downstream task – domain data selection. The multi-domain corpus proposed in Koehn & Knowles (2017) includes parallel text in German and English from five domains: translations of the Koran, subtitles, medical, legal and IT-related text, available via OPUS (Aulamo & Tiedemann (2019)), split as Aharoni & Goldberg (2020).

We encode multi-domain data at the sentence level into vector representations by pre-trained models. (i) MLM-based model: BERT (Devlin et al. (2018)), DistilBERT (Sanh et al. (2019)) and RoBERTa (Liu et al. (2019)), in both the base and large versions; (ii) autoregressive model: GPT-2 (Radford et al. (2018)) and XLNet (Yang et al. (2019)); (iii) baseline model: word2vex (Mikolov et al. (2013)). In all cases we use the implementations from the HuggingFace Transformers toolkit (Wolf et al.

| | Pre-trained Model | without PCA | PCA | RK | RK-Spat | BS | BS-Spat |
|---|---|---|---|---|---|---|---|
| | word2vec | 45.93 | 54.37 | **58.56** | 49.76 | 52.24 | 49.16 |
| | BERT-base | 85.81 | 78.98 | **87.45** | 87.27 | 86.13 | 86.42 |
| | BERT-large | 72.25 | **88.00** | 87.91 | 87.59 | 72.54 | 86.89 |
| GMM-5 | DistilBERT | 73.35 | **87.53** | 87.03 | 86.84 | 85.83 | 85.94 |
| | RoBERTa-base | 70.21 | 78.95 | 79.02 | **79.08** | 72.79 | 61.86 |
| | RoBERTa-large | 59.32 | 73.74 | 80.94 | 73.51 | 73.38 | **86.15** |
| | GPT-2 | 37.82 | **70.25** | 68.97 | 68.96 | 69.42 | 69.47 |
| | XLNet | 30.35 | 56.31 | 56.86 | **56.92** | 51.29 | 51.19 |
| | word2vec | 65.80 | **71.13** | 66.43 | 68.29 | 63.76 | 66.44 |
| | BERT-base | 84.65 | **88.76** | 88.50 | 88.45 | 86.34 | 87.71 |
| | BERT-large | 84.32 | **88.21** | 86.71 | 86.83 | 86.24 | 87.55 |
| GMM-10 | DistilBERT | 86.00 | 83.17 | 86.16 | **87.62** | 84.22 | 85.12 |
| | RoBERTa-base | 79.26 | **88.72** | 86.84 | 86.32 | 84.51 | 86.06 |
| | RoBERTa-large | 81.17 | 88.13 | 89.27 | 89.21 | 88.67 | **89.29** |
| | GPT-2 | 40.26 | **84.34** | 83.47 | 83.09 | 83.50 | 83.36 |
| | XLNet | 36.57 | **70.47** | 68.92 | 68.37 | 64.43 | 55.64 |

Table 1: Clustering Purity for unsupervised domain clustering. Best results in each row are marked in bold. Explanatory variables involved in GMM-$k$ are set to be original data without PCA, standard PCA and FPC scores computed from RK(-Spat) & BS(-Spat) sparsified covariance estimators.

(2019)). We cluster these vector representations through a Gaussian Mixture Model (GMM-$k$) where $k$ is the number of predetermined clusters. The estimated FPC scores are naturally independent random variables and converge to the infeasible true FPC scores at the rate of $n^{-1/2}$ in probability, see Theorem 5 (iii). It is reasonable to set the estimated FPC scores as explanatory variables in GMM-$k$ step, which are computed from $\hat{\xi}_{ik} = \hat{\lambda}_k^{-1/2} \int \{h_i(t) - \hat{m}(t)\} \hat{\psi}_k(t) \, dt$ with $\hat{\lambda}_k$ and $\hat{\psi}_k$ the estimates of eigenvalues and eigenfunctions of covariance estimator. We also set original vectors and FPC scores computed from standard PCA method as explanatory variables for comparison. To evaluate whether the resulting clusters indeed capture the domains, we measure the Clustering Purity, which is a well-known metric for evaluating clustering, see Schütze et al. (2008).

Table 1 shows MLM-based models dominate over word2vec and auto-regressive models. Mainly because MLM-based models use the entire sentence context when generating the representations, while auto-regressive models only use the past context, and word2vec uses a limited window context. Direct modeling with data without dimensionality reduction has the worst results, and PCA significantly improves the performance in all cases. The last four columns of Table 1 reflect that Purity of the domain-clustering task is not sacrificed and even slightly improves in some cases. Hence, modeling on FPC scores computed from sparsified covariance estimators are shown to be efficient and effective. In sum, by applying our sparsification methods and related covariance estimation, we achieve the same performance as standard PCA on the basis of accelerating computation speed.

## 5 CONCLUSIONS AND LIMITATION

In this paper, RK (RK-Spat) estimator converges to an averaged sample estimator without sparsification at the rate of $\mathcal{O}_p(n^{-1/2})$, and BS (BS-Spat) estimator converges at the rate of $\mathcal{O}_p(n^{-1/2})$. We further characterize the uniform weak convergence of the corresponding estimation of eigenvalues and eigenvectors. It is necessary to take spatial factor into account when correlation across nodes is non-negligible, thus standard approach to averaging sample vectors can lead to high estimation error. And spline interpolation is carried out to avoid the loss of overall data information. Theoretical results are backed by simulation and application.

A few more issues still merit further research. The AIC selection method works well in practice, but a stronger theoretical justification for its use is still needed. Our work focuses on the approximation and estimation, while in recent years, there has been a great deal of work on deriving approximate distribution, which is crucial for making global inference. It is also worth exploring to extend our novel sparsification methodology to functional regression model and large-scale longitudinal model, which is expected to find more applications in various scientific fields. Covariance estimation in such models is a significant challenge and requires more in-depth investigation.

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

## A    DECOMPOSITION

For estimation in FIXED-SPARSIFICATION method, we define the design matrix for B-spline regression as

$$\mathbf{B} = \{\mathbf{B}(1/d), \ldots, \mathbf{B}(d/d)\}^\top = \begin{pmatrix} B_{1,p}(1/d) & \cdots & B_{J_s+p,p}(1/d) \\ \vdots & \ldots & \vdots \\ B_{1,p}(d/d) & \cdots & B_{J_s+p,p}(d/d) \end{pmatrix}.$$

Denote by $\mathbf{V}_{n,p}$ the empirical inner product matrix of B-spline basis $\{B_{\ell,p}(t)\}_{\ell=1}^{J_s+p}$, i.e.

$$\mathbf{V}_{n,p} = \left\{\langle B_{\ell,p}, B_{\ell',p}\rangle_d\right\}_{\ell,\ell'=1}^{J_s+p} = d^{-1}\mathbf{B}^\top\mathbf{B}.$$

Denote $\boldsymbol{x}_i = (x_i(1/d),\ldots,x_i(d/d))^\top$, $\boldsymbol{m} = (m(1/d),\ldots,m(d/d))^\top$ and $\boldsymbol{Z}_i = (Z_i(1/d),\ldots,Z_i(d/d))^\top$. The form of model (1) ensures that spline estimator $h_i(\cdot)$ allows representation $h_i(\cdot) = d^{-1}\mathbf{B}(\cdot)^\top\mathbf{V}_{n,p}^{-1}\mathbf{B}^\top\boldsymbol{x}_i = \hat{m}(\cdot) + \hat{Z}_i(\cdot)$, where

$$\hat{m}(\cdot) = d^{-1}\mathbf{B}(\cdot)^\top\mathbf{V}_{n,p}^{-1}\mathbf{B}^\top\boldsymbol{m},$$
$$\hat{Z}_i(\cdot) = d^{-1}\mathbf{B}(\cdot)^\top\mathbf{V}_{n,p}^{-1}\mathbf{B}^\top\boldsymbol{Z}_i.$$

## B PROOF OF THEOREM 1

According to the definition of $\hat{G}$ and $\bar{G}$, the MSE can be computed as

$$\mathbb{E}\|\hat{G} - \bar{G}\|^2 = \sum_{j,j'=1}^d \mathbb{E}\left(\|\frac{1}{n}\left(\frac{d}{J_s}\right)^2\sum_{i=1}^n\left(h_{ij} - \bar{h}_j\right)\left(h_{ij'} - \bar{h}_{j'}\right) - \frac{1}{n}\sum_{i=1}^n\left(x_{ij} - \bar{m}_j\right)\left(x_{ij'} - \bar{m}_{j'}\right)\|^2\right)$$

Now, as $\mathbb{E}\left(\frac{d}{J_s}h_{ij}\right) = x_{ij}$ and $\mathbb{E}\left(\frac{d}{J_s}h_{ij'}\right) = x_{ij'}$, it holds that

$$\frac{1}{n^2}\sum_{i=1}^n\mathbb{E}\left(\left(\frac{d}{J_s}\right)^2\left(h_{ij} - \bar{h}_j\right)\left(h_{ij'} - \bar{h}_{j'}\right) - \left(x_{ij} - \bar{m}_j\right)\left(x_{ij'} - \bar{m}_{j'}\right)\right)^2$$

$$= \frac{1}{n^2}\sum_{i=1}^n\left(\mathbb{E}\left(\left(\frac{d}{J_s}\right)^2\left(h_{ij} - \bar{h}_j\right)\left(h_{ij'} - \bar{h}_{j'}\right)\right)^2 - \mathbb{E}\left(\left(x_{ij} - \bar{m}_j\right)\left(x_{ij'} - \bar{m}_{j'}\right)\right)^2\right)$$

where $\bar{h}_j$ stands for the $j$-th element of the $d$-dimensional vector $\bar{h}$. Since $h_{ij} = x_{ij}$ with probability $J_s/d$ and $h_{ij} = 0$ with probability $1 - J_s/d$, therefore

$$\mathbb{E}\left(\left(\frac{d}{J_s}\right)^2\left(h_{ij} - \bar{h}_j\right)\left(h_{ij'} - \bar{h}_{j'}\right)\right)^2 = \left(\frac{d}{J_s}\right)^4\mathbb{E}\left(\left(h_{ij} - \bar{h}_j\right)\left(h_{ij'} - \bar{h}_{j'}\right)\right)^2$$

$$= \left(\frac{d}{J_s}\right)^2\mathbb{E}\left(\left(x_{ij} - \bar{m}_j\right)\left(x_{ij'} - \bar{m}_{j'}\right)\right)^2$$

Since elements in each vector are assumed to be generated independently for RANDOM-SPARSIFICATION, that is $x_{ij}$ is independence of $x_{ij'}$, $1 \le i \le n$, $j \ne j'$. Hence,

$$\mathbb{E}\|\hat{G} - \bar{G}\|^2 = \frac{1}{n^2}\left(\left(\frac{d}{J_s}\right)^2 - 1\right)\sum_{i=1}^n\sum_{j,j'=1}^d\mathbb{E}\left(\left(x_{ij} - \bar{m}_j\right)\left(x_{ij'} - \bar{m}_{j'}\right)\right)^2$$

$$= \frac{1}{n^2}\left(\left(\frac{d}{J_s}\right)^2 - 1\right)\sum_{i=1}^n\mathbb{E}\left(\sum_{j=1}^d\left(x_{ij} - \bar{m}_j\right)^2\sum_{j'=1}^d\left(x_{ij'} - \bar{m}_{j'}\right)^2\right)$$

$$= \frac{1}{n^2}\left(\left(\frac{d}{J_s}\right)^2 - 1\right)R_1$$

where $R_1 = \sum_{i=1}^n\|x_i - \bar{m}\|^4$.

Moreover, according to Assumption 6,

$$n^{-1}\left(\left(\frac{d}{J_s}\right)^2 - 1\right) \asymp d^{-\theta}\left(\frac{d}{d^\gamma C_d}\right)^2 \to 0.$$

The bounded $R_1$ further tells that $\mathbb{E}\|\hat{G} - \bar{G}\|^2 = O(n^{-1})$, and consequently $\|\hat{G} - \bar{G}\| = O_p(n^{-1/2})$.

## C  PROOF OF PROPOSITION 1

The trick is to introduce the random variable $\xi_{ij}$ to aid in the computation of conditional expectations $\mathbb{E}_{M_j|M_j\geq 1}(\cdot)$. Let $\xi_{ij}$ be an indicator random variable which takes value in $\{0,1\}$, depending on whether $h_{ij} = x_{ij}$ or not for $1 \leq i \leq n$, $1 \leq j \leq d$, leading to $\mathbb{E}_{M_j|M_j\geq 1, M_{j'}\geq 1}(\cdot) = \mathbb{E}_{M_j|\xi_{ij}=1,\xi_{ij'}=1}(\cdot)$, $\mathbb{E}_{M_j|M_j=0, M_{j'}\geq 1}(\cdot) = \mathbb{E}_{M_j|\xi_{ij}=0,\xi_{ij'}=1}(\cdot)$, $\mathbb{E}_{M_j|M_j=0, M_{j'}=0}(\cdot) = \mathbb{E}_{M_j|\xi_{ij}=0,\xi_{ij'}=0}(\cdot)$. Event $\{\xi_{ij}=1,\xi_{ij'}=1\}$ happens with probability $\left(\frac{J_s}{d}\right)^2$, event $\{\xi_{ij}=0,\xi_{ij'}=1\}$ happens with probability $\left(\frac{J_s}{d}\right)\left(1-\frac{J_s}{d}\right)$ and event $\{\xi_{ij}=0,\xi_{ij'}=0\}$ happens with probability $\left(1-\frac{J_s}{d}\right)^2$.

Case 1: With probability $\left(1-\frac{J_s}{d}\right)^2$, event $\{\xi_{ij}=0,\xi_{ij'}=0\}$ holds, which implies $h_{ij}=0$ and $h_{ij'}=0$. Therefore,

$$\mathbb{E}_{M_j,M_{j'}|\xi_{ij}=0,\xi_{ij'}=0}\left(\frac{\bar{\beta}^2\left(h_{ij}-\bar{h}_j\right)\left(h_{ij'}-\bar{h}_{j'}\right)}{T\left(M_j\right)T\left(M_{j'}\right)}\right)$$
$$=\mathbb{E}_{M_j|\xi_{ij}=0}\left(\frac{\bar{\beta}\left(h_{ij}-\bar{h}_j\right)}{T\left(M_j\right)}\right)\mathbb{E}_{M_{j'}|\xi_{ij'}=0}\left(\frac{\bar{\beta}\left(h_{ij'}-\bar{h}_{j'}\right)}{T\left(M_{j'}\right)}\right)=0.$$

Case 2: With probability $\left(\frac{J_s}{d}\right)\left(1-\frac{J_s}{d}\right)$, event $\{\xi_{ij}=0,\xi_{ij'}=1\}$ holds, which implies $h_{ij}=0$ and $h_{ij'}=x_{ij'}$. Still we have

$$\mathbb{E}_{M_j,M_{j'}|\xi_{ij}=0,\xi_{ij'}=1}\left(\frac{\bar{\beta}^2\left(h_{ij}-\bar{h}_j\right)\left(h_{ij'}-\bar{h}_{j'}\right)}{T\left(M_j\right)T\left(M_{j'}\right)}\right)=0.$$

Case 3: With probability $\left(\frac{J_s}{d}\right)^2$, event $\{\xi_{ij}=1,\xi'=1\}$ holds, which implies $h_{ij}=x_{ij}$ and $h_{ij'}=x_{ij'}$. Therefore,

$$\mathbb{E}_{M_j,M_{j'}|\xi_{ij}=1,\xi_{ij'}=1}\left(\frac{\bar{\beta}^2\left(h_{ij}-\bar{h}_j\right)\left(h_{ij'}-\bar{h}_{j'}\right)}{T\left(M_j\right)T\left(M_{j'}\right)}\right)$$
$$=\mathbb{E}_{M_j,M_{j'}|M_j\geq 1,M_{j'}\geq 1}\left(\frac{\bar{\beta}^2\left(h_{ij}-\bar{h}_j\right)\left(h_{ij'}-\bar{h}_{j'}\right)}{T\left(M_j\right)T\left(M_{j'}\right)}\right)$$
$$=\bar{\beta}^2\left(x_{ij}-\bar{m}_j\right)\left(x_{ij'}-\bar{m}_{j'}\right)\mathbb{E}_{M_j,M_{j'}|M_j\geq 1,M_{j'}\geq 1}\left(\frac{1}{T\left(M_j\right)T\left(M_{j'}\right)}\right).$$

A crucial observation is that $\xi_{ij}=1$ only implies $M_j\geq 1$ and does not give any other information about $M_j$. Taking expectation with respect to $\xi_{ij}$ we have,

$$\mathbb{E}_{\xi_{ij},\xi_{ij'}}\mathbb{E}_{M_j,M_{j'}|\xi_{ij},\xi_{ij'}}\left(\frac{\bar{\beta}^2\left(h_{ij}-\bar{h}_j\right)\left(h_{ij'}-\bar{h}_{j'}\right)}{T\left(M_j\right)T\left(M_{j'}\right)}\right)$$
$$=\left(\frac{J_s}{d}\right)^2\bar{\beta}^2\left(x_{ij}-\bar{m}_j\right)\left(x_{ij'}-\bar{m}_{j'}\right)\mathbb{E}_{M_j,M_{j'}|M_j\geq 1,M_{j'}\geq 1}\left(\frac{1}{T\left(M_j\right)T\left(M_{j'}\right)}\right)$$
$$=\left(x_{ij}-\bar{m}_j\right)\left(x_{ij'}-\bar{m}_{j'}\right).$$

This proves Proposition 1.

## D    PROOF OF THEOREM 2

According to the definition, the $(j, j')$-th element of $\hat{G}^{\text{SPAT}}$ is

$$
\hat{G}^{\text{SPAT}}_{jj'} = \frac{1}{n} \left(\frac{d}{J_s}\right)^2 \frac{\left(\mathbb{E}_{M_j|M_j \geq 1}\left(\frac{1}{T(M_j)}\right)\mathbb{E}_{M_{j'}|M_{j'} \geq 1}\left(\frac{1}{T(M_{j'})}\right)\right)^{-1}}{T(M_j)\,T(M_{j'})} \sum_{i=1}^{n} \left(h_{ij} - \bar{h}_j\right)\left(h_{ij'} - \bar{h}_{j'}\right)
$$

$$
= \frac{1}{n} \frac{\bar{\beta}^2}{T(M_j)\,T(M_{j'})} \sum_{i=1}^{n} \left(h_{ij} - \bar{h}_j\right)\left(h_{ij'} - \bar{h}_{j'}\right)
$$

MSE can be computed as

$$
\mathbb{E}\|\hat{G}^{\text{SPAT}} - \bar{G}\|^2 = \sum_{j,j'=1}^{d} \mathbb{E}\left(\hat{G}^{\text{SPAT}}_{jj'} - \bar{G}_{jj'}\right)^2
$$

$$
= \sum_{j,j'=1}^{d} \mathbb{E}\left(\frac{1}{n}\frac{\bar{\beta}^2}{T(M_j)\,T(M_{j'})}\sum_{i=1}^{n}\left(h_{ij} - \bar{h}_j\right)\left(h_{ij'} - \bar{h}_{j'}\right) - \frac{1}{n}\sum_{i=1}^{n}\left(x_{ij} - \bar{m}_j\right)\left(x_{ij'} - \bar{m}_{j'}\right)\right)^2
$$

$$
\tag{5}
$$

As the estimator is designed to be unbiased, i.e.,

$$
\mathbb{E}\left(\frac{1}{n}\frac{\bar{\beta}^2}{T(M_j)\,T(M_{j'})}\sum_{i=1}^{n}\left(h_{ij} - \bar{h}_j\right)\left(h_{ij'} - \bar{h}_{j'}\right)\right) = \frac{1}{n}\sum_{i=1}^{n}\left(x_{ij} - \bar{m}_j\right)\left(x_{ij'} - \bar{m}_{j'}\right),
$$

it holds that

$$
\mathbb{E}\left(\frac{1}{n}\frac{\bar{\beta}^2}{T(M_j)\,T(M_{j'})}\sum_{i=1}^{n}\left(h_{ij} - \bar{h}_j\right)\left(h_{ij'} - \bar{h}_{j'}\right) - \frac{1}{n}\sum_{i=1}^{n}\left(x_{ij} - \bar{m}_j\right)\left(x_{ij'} - \bar{m}_{j'}\right)\right)^2
$$

$$
= \frac{1}{n^2}\mathbb{E}\left(\frac{\bar{\beta}^2}{T(M_j)\,T(M_{j'})}\sum_{i=1}^{n}\left(h_{ij} - \bar{h}_j\right)\left(h_{ij'} - \bar{h}_{j'}\right)\right)^2 - \frac{1}{n^2}\left(\sum_{i=1}^{n}\left(x_{ij} - \bar{m}_j\right)\left(x_{ij'} - \bar{m}_{j'}\right)\right)^2
$$

$$
\tag{6}
$$

We now analyze the first term above.

$$
\mathbb{E}\left(\frac{\bar{\beta}^2}{T(M_j)\,T(M_{j'})}\sum_{i=1}^{n}\left(h_{ij} - \bar{h}_j\right)\left(h_{ij'} - \bar{h}_{j'}\right)\right)^2
$$

$$
= \sum_{i=1}^{n} \bar{\beta}^4 \mathbb{E}\left(\frac{\left(h_{ij} - \bar{h}_j\right)^2\left(h_{ij'} - \bar{h}_{j'}\right)^2}{T(M_j)^2\,T(M_{j'})^2}\right)
$$

$$
+ 2\sum_{i=1}^{n}\sum_{k=i+1}^{n}\bar{\beta}^4 \mathbb{E}\left(\frac{\left(h_{ij} - \bar{h}_j\right)\left(h_{kj} - \bar{h}_j\right)\left(h_{ij'} - \bar{h}_{j'}\right)\left(h_{kj'} - \bar{h}_{j'}\right)}{T(M_j)^2\,T(M_{j'})^2}\right). \tag{7}
$$

Note that the expectation is taken over the randomness in $h_{ij}$ as well as $T(M_j)$. Further, $\bar{\beta}^4 \frac{\left(h_{ij}-\bar{h}_j\right)^2\left(h_{ij'}-\bar{h}_{j'}\right)^2}{T(M_j)^2 T(M_{j'})^2}$ is non-zero only when a node $i$ samples coordinate $j$ and $j'$, i.e., $h_{ij} = x_{ij}$ and $h_{ij'} = x_{ij'}$. This implies that $M_j \geq 1$ and $M_{j'} \geq 1$. Therefore, by the law of total expectation,

we have

$$
\bar{\beta}^4 \mathbb{E}\left( \frac{\left(h_{ij} - \bar{h}_j\right)^2 \left(h_{ij'} - \bar{h}_{j'}\right)^2}{T\left(M_j\right)^2 T\left(M_{j'}\right)^2} \right)
$$

$$
= \bar{\beta}^4 \mathbb{E}_{M_j | M_j \geq 1}\left( \frac{J_s \left(x_{ij} - \bar{m}_j\right)^2}{dT\left(M_j\right)^2} \right) \mathbb{E}_{M_{j'} | M_{j'} \geq 1}\left( \frac{J_s \left(x_{ij'} - \bar{m}_{j'}\right)^2}{dT\left(M_{j'}\right)^2} \right)
$$

$$
= \left( \bar{\beta}^4 \sum_{r,r'=1}^{n} \frac{J_s}{dT(r)^2} \frac{J_s}{dT(r')^2} \binom{n-1}{r-1}\binom{n-1}{r'-1} \left(\frac{J_s}{d}\right)^{r+r'-2} \left(1 - \frac{J_s}{d}\right)^{2n-r-r'} \right)
$$

$$
\times \left(x_{ij} - \bar{m}_j\right)^2 \left(x_{ij'} - \bar{m}_{j'}\right)^2
$$

$$
= \left(\frac{d}{J_s} + c_1\right)^2 \left(x_{ij} - \bar{m}_j\right)^2 \left(x_{ij'} - \bar{m}_{j'}\right)^2 \tag{8}
$$

where $c_1$ is defined in Theorem 2.

Following a similar argument as above, note that $\frac{\left(h_{ij} - \bar{h}_j\right)\left(h_{kj} - \bar{h}_j\right)\left(h_{ij'} - \bar{h}_{j'}\right)\left(h_{kj'} - \bar{h}_{j'}\right)}{T(M_j)^2 T\left(M_{j'}\right)^2}$ is non-zero only when nodes $i$ and $k$ sample coordinate $j$ and $j'$, i.e., $h_{ij} = x_{ij}$, $h_{kj} = x_{kj}$, $h_{ij'} = x_{ij'}$, $h_{kj'} = x_{kj'}$. This implies that $M_j \geq 2$ and $M_{j'} \geq 2$. Therefore, by the law of total expectation, we have

$$
\bar{\beta}^4 \mathbb{E}\left( \frac{\left(h_{ij} - \bar{h}_j\right)\left(h_{kj} - \bar{h}_j\right)\left(h_{ij'} - \bar{h}_{j'}\right)\left(h_{kj'} - \bar{h}_{j'}\right)}{T\left(M_j\right)^2 T\left(M_{j'}\right)^2} \right)
$$

$$
= \bar{\beta}^4 \mathbb{E}_{M_j | M_j \geq 2}\left( \frac{J_s \left(x_{ij} - \bar{m}_j\right)\left(x_{kj} - \bar{m}_j\right)}{dT\left(M_j\right)^2} \right) \mathbb{E}_{M_{j'} | M_{j'} \geq 2}\left( \frac{J_s \left(x_{ij'} - \bar{m}_{j'}\right)\left(x_{kj'} - \bar{m}_{j'}\right)}{dT\left(M_{j'}\right)^2} \right)
$$

$$
= \left( \bar{\beta}^4 \sum_{r,r'=2}^{n} \frac{J_s}{dT\left(r'\right)^2} \frac{J_s^2}{d^2 T\left(r'\right)^2} \binom{n-2}{r-2}\binom{n-2}{r'-2} \left(\frac{J_s^2}{d^2}\right)^{r+r'-4} \left(1 - \frac{J_s}{d}\right)^{2n-r-r'} \right)
$$

$$
\times \left(x_{ij} - \bar{m}_j\right)\left(x_{kj} - \bar{m}_j\right)\left(x_{ij'} - \bar{m}_{j'}\right)\left(x_{kj'} - \bar{m}_{j'}\right)
$$

$$
= \left(1 - c_2\right)^2 \left(x_{ij} - \bar{m}_j\right)\left(x_{kj} - \bar{m}_j\right)\left(x_{ij'} - \bar{m}_{j'}\right)\left(x_{kj'} - \bar{m}_{j'}\right) \tag{9}
$$

where $c_2$ is defined in Theorem 2. Substituting (8) and (9) into (7), we get

$$
\mathbb{E}\left( \frac{\bar{\beta}^2}{T\left(M_j\right)T\left(M_{j'}\right)} \sum_{i=1}^{n} \left(h_{ij} - \bar{h}_j\right)\left(h_{ij'} - \bar{h}_{j'}\right) \right)^2
$$

$$
= \left(\frac{d}{J_s} + c_1\right)^2 \sum_{i=1}^{n} \left(x_{ij} - \bar{m}_j\right)^2 \left(x_{ij'} - \bar{m}_{j'}\right)^2
$$

$$
+ \left(1 - c_2\right)^2 \sum_{i=1}^{n} \sum_{k=i+1}^{n} \left(x_{ij} - \bar{m}_j\right)\left(x_{kj} - \bar{m}_j\right)\left(x_{ij'} - \bar{m}_{j'}\right)\left(x_{kj'} - \bar{m}_{j'}\right) \tag{10}
$$

Now, substituting (10) into (6), we get

$$
\mathbb{E}\left( \frac{1}{n}\frac{\bar{\beta}^2}{T\left(M_j\right)T\left(M_{j'}\right)} \sum_{i=1}^{n} \left(h_{ij} - \bar{h}_j\right)\left(h_{ij'} - \bar{h}_{j'}\right) - \frac{1}{n}\sum_{i=1}^{n} \left(x_{ij} - \bar{m}_j\right)\left(x_{ij'} - \bar{m}_{j'}\right) \right)^2
$$

$$
= \frac{1}{n^2}\left( \left(\frac{d}{J_s} + c_1\right)^2 - 1 \right) \sum_{i=1}^{n} \left(x_{ij} - \bar{m}_j\right)^2 \left(x_{ij'} - \bar{m}_{j'}\right)^2
$$

$$
+ \frac{1}{n^2}\left( \left(1 - c_2\right)^2 - 1 \right) \sum_{i=1}^{n} \sum_{k=i+1}^{n} \left(x_{ij} - \bar{m}_j\right)\left(x_{kj} - \bar{m}_j\right)\left(x_{ij'} - \bar{m}_{j'}\right)\left(x_{kj'} - \bar{m}_{j'}\right) \tag{11}
$$

Finally substituting (11) into (5) we get,

$$\mathbb{E}\|\hat{G}^{\text{SPAT}} - \bar{G}\|^2 = \frac{1}{n^2}\left(\left(\frac{d}{J_s} + c_1\right)^2 - 1\right)R_1 + \frac{1}{n^2}\left((1 - c_2)^2 - 1\right)R_2$$

where $R_1 = \sum_{i=1}^n \|x_i - \bar{m}\|^4$ and $R_2 = 2\sum_i^n \sum_{k=i+1}^n \left\langle (x_i - \bar{m})^2, (x_k - \bar{m})^2 \right\rangle$.

Moreover, according to Assumption 6 and the boundness of $c_1$,

$$n^{-1}\left(\left(\frac{d}{J_s} + c_1\right)^2 - 1\right) \asymp d^{-\theta}\left(\frac{d}{d^\gamma C_d}\right)^2 + d^{-\theta}\left(\frac{d}{d^\gamma C_d}\right) \to 0.$$

The bounded $R_1$ further tells that $n^{-2}\left(\left(\frac{d}{J_s} + c_1\right)^2 - 1\right)R_1 = O(n^{-1})$. The bounded $c_2$ and $R_2$ assure that $n^{-2}\left((1 - c_2)^2 - 1\right)R_2 = O(n^{-2})$. Therefore, $\mathbb{E}\|\hat{G}^{\text{SPAT}} - \bar{G}\|^2 = O(n^{-1})$ and thus $\|\hat{G}^{\text{SPAT}} - \bar{G}\| = O_p(n^{-1/2})$.

# E    PROOF OF THEOREM 3

Note that $c_1$ and $c_2$ are completely determined by the original data, while $R_1$ and $R_2$ are dependent on function $T(\cdot)$. Observing the result in Theorem 2, the only term that depends on $T(\cdot)$ is $\left(\frac{d}{J_s} + c_1\right)^2 R_1 + (1 - c_2)^2 R_2$ since these terms contain $c_1$ and $c_2$, which are computed from $T(\cdot)$. Thus to find the function $T^*(\cdot)$ that minimizes the MSE, we just need to minimize this term.

From the definitions of $c_1$ and $c_2$ in Theorem 2, we can obtain the following expression for $T^*(\cdot)$

$$T^*(r) = \underset{T}{\arg\min}\left\{\bar{\beta}^4\left(\sum_{r=1}^n \frac{J_s}{dT(r)^2}\binom{n-1}{r-1}\left(\frac{J_s}{d}\right)^{r-1}\left(1 - \frac{J_s}{d}\right)^{n-r}\right)^2 \right.$$
$$\left. + \frac{R_2}{R_1}\bar{\beta}^4\left(\sum_{r=2}^n \frac{J_s^2}{d^2 T(r)^2}\binom{n-2}{r-2}\left(\frac{J_s}{d}\right)^{r-2}\left(1 - \frac{J_s}{d}\right)^{n-r}\right)^2\right\}. \quad (12)$$

We claim that $T^*(r) = \left(1 + \frac{R_2}{R_1}\left(\frac{r-1}{n-1}\right)^2\right)^{1/2}$ is an optimal solution for our objective defined in (12). To see this, consider the following cases,

Case 1: $p = 0$ or $p = 1$. In this case $c_1$ and $c_2$ are independent of $T(\cdot)$ and hence our objective does not depend on the choice of $T(\cdot)$.

Case 2: $0 < p < 1$, we define

$$\mathbf{w}^* = \underset{\mathbf{w}}{\arg\min}\frac{\mathbf{w}^\top \mathbf{A}\mathbf{w}}{(\mathbf{b}^\top \mathbf{w})^2}, \quad (13)$$

where $\mathbf{w}$ is a $n$-dimensional vector whose $r$-th entry is $w_r = 1/T(r)^2$, $\mathbf{b}$ is a vector whose $r$-th entry is

$$b_r = \left(\binom{n-1}{r-1}p^{r-1}(1-p)^{n-r}\right)^2$$

where $p = \frac{J_s}{d}$, and $\mathbf{A}$ is a diagonal matrix whose $r$-th diagonal entry is

$$A_{rr} = \left(\binom{n-1}{r-1}p^{r-1}(1-p)^{n-r}\right)^2 + \frac{R_2}{R_1}\left(p\binom{n-2}{r-2}p^{r-2}(1-p)^{n-r}\right)^2$$
$$= b_r\left(1 + \frac{R_2}{R_1}\left(\frac{r-1}{n-1}\right)^2\right).$$

Note that $A_{rr} > 0$ for all $r \in \{1, 2, \ldots, n\}$ which implies that $\mathbf{w} \rightarrow \mathbf{A}^{1/2}\mathbf{w}$ is a one-to-one mapping. Therefore setting $\mathbf{z} = \mathbf{A}^{1/2}\mathbf{w}$, the objective in (13) reduces to

$$\mathbf{z}^* = \arg\min_{\mathbf{z}} \frac{\|\mathbf{z}\|^2}{\left(\mathbf{b}^\top \mathbf{A}^{-1/2}\mathbf{z}\right)^2} \tag{14}$$

Observe that the objectives (13) and (14) are invariant to the scale of $T(\cdot)$, $\mathbf{w}$ and $\mathbf{z}$ respectively and thus the solutions will be unique up to a scaling factor. Therefore, in the case of (14), it is sufficient to solve the reduced objective,

$$\mathbf{z}^* = \arg\min_{\mathbf{z}, \|\mathbf{z}\|=1} \frac{\|\mathbf{z}\|^2}{\left(\mathbf{b}^\top \mathbf{A}^{-1/2}\mathbf{z}\right)^2} = \arg\min_{\mathbf{z}, \|\mathbf{z}\|=1} \frac{1}{\left(\mathbf{b}^\top \mathbf{A}^{-1/2}\mathbf{z}\right)^2}$$

which is minimized by $\mathbf{z}^* = \frac{\mathbf{A}^{-1/2}\mathbf{b}}{\|\mathbf{A}^{-1/2}\mathbf{b}\|}$. Therefore, the optimal solution (up to a constant) is $\mathbf{w}^* = \mathbf{A}^{-1/2}\left(\mathbf{A}^{-1/2}\mathbf{b}\right)$. Correspondingly, we conclude that

$$T^*(r) = (w_r^*)^{-1/2} = \left(\frac{A_{rr}}{b_r}\right)^{1/2} = \left(1 + \frac{R_2}{R_1}\left(\frac{r-1}{n-1}\right)^2\right)^{1/2}.$$

minimizes (12), and consequently minimizes the MSE of the RK-SPAT estimator.

## F    PROOF OF THEOREM 4

PROOF OF (I) We first provide several technical lemmas.

**Lemma 1.** *Let $W_i \sim N\left(0, \sigma_i^2\right)$, $\sigma_i > 0$, $i \in \{1, 2, \ldots, n\}$, then for $n > 2$, $a > 2$,*

$$\mathbb{P}\left(\max_{1 \leq i \leq n} |W_i/\sigma_i| > a\sqrt{\log n}\right) < \sqrt{\frac{2}{\pi}} n^{1-a^2/2}.$$

*Hence, $(\max_{1 \leq i \leq n} |W_i|) / (\max_{1 \leq i \leq n} \sigma_i) \leq \max_{1 \leq i \leq n} |W_i/\sigma_i| = \mathcal{O}_{a.s.}(\sqrt{\log n})$.*

PROOF. Note that for $n > 2, a > 2$,

$$\mathbb{P}\left(\max_{1 \leq i \leq n}\left|\frac{W_i}{\sigma_i}\right| > a\sqrt{\log n}\right) \leq \sum_{i=1}^{n} \mathbb{P}\left(\left|\frac{W_i}{\sigma_i}\right| > a\sqrt{\log n}\right)$$

$$\leq 2n\{1 - \Phi(a\sqrt{\log n})\} < 2n\frac{\phi(a\sqrt{\log n})}{a\sqrt{\log n}} \leq 2n\phi(a\sqrt{\log n}) = \sqrt{\frac{2}{\pi}} n^{1-a^2/2}$$

where $\phi(x)$ denotes standard probability density function at $x$ and $\Phi(x)$ denotes the corresponding cumulative distribution function. Lemma 1 follows by applying Borel-Cantelli lemma.

**Lemma 2.** *As $n \rightarrow \infty$, we have*

$$\max_{1 \leq i \leq n} \|h_i - x_i\|_\infty = \mathcal{O}_{a.s.}\left\{J_s^{-p^*}(n\log n)^{2/r_0}\right\} = o_{a.s.}\left(n^{-1/2}\right).$$

PROOF. The trajectory $x_i(t)$ can be written as $x_i(t) = m(t) + \sum_{k=1}^{\infty} \xi_{ik}\phi_k(t)$. Denote $\phi_k = (\phi_k(1/d), \ldots, \phi_k(d/d))^\top$, and let $\hat{\phi}_k(t) = d^{-1}\mathbf{B}(t)^\top \mathbf{V}_{n,p}^{-1}\mathbf{B}^\top \phi_k$ be the B-spline smoothing of $\phi_k(t)$. The linearity of spline smoothing implies that

$$h_i(t) - x_i(t) = \hat{m}(t) - m(t) + \sum_{k=1}^{\infty} \xi_{ik}\left\{\hat{\phi}_k(t) - \phi_k(t)\right\}.$$

Lemma A.4 in Cao et al. (2012) assures there exists a constant $C_{q,\mu} > 0$, such that

$$\|\hat{m} - m\|_\infty \leq C_{q,\mu}\|m\|_{q,\mu} J_s^{-p^*}, \tag{15}$$

$$\left\|\hat{\phi}_k - \phi_k\right\|_\infty \leq C_{q,\mu}\|\phi_k\|_{q,\mu} J_s^{-p^*}, \quad k \geq 1 \tag{16}$$

Thus, with norm inequality, we have

$$\|h_i - x_i\|_\infty \le \|\hat{m} - m\|_\infty + \sum_{k=1}^\infty |\xi_{ik}| \left\|\hat{\phi}_k - \phi_k\right\|_\infty \le C_{q,\mu} W_i J_s^{-p^*}$$

where $W_i = \|m\|_{q,\mu} + \sum_{k=1}^\infty |\xi_{ik}| \|\phi_k\|_{q,\mu}$ are i.i.d. nonnegative random variables. $W_i^{r_0}$ has a finite absolute moment and we have

$$\mathbb{P}\left\{\max_{1\le i\le n} W_i > (n\log n)^{2/r_0}\right\} \le n\frac{\mathbb{E}W_i^{r_0}}{(n\log n)^2} = \mathbb{E}W_i^{r_0}(n\log n)^{-2}$$

which implies

$$\sum_{n=1}^\infty \mathbb{P}\left\{\max_{1\le i\le n} W_i > (n\log n)^{2/r_0}\right\} \le \mathbb{E}W_i^{r_0} \sum_{n=1}^\infty (n\log n)^{-2} < +\infty$$

According to Borel Cantelli lemma, we conclude that $\max_{1\le i\le n} W_i = \mathcal{O}_{a.s.}\left\{(n\log n)^{2/r_0}\right\}$, which together with (15) and (16), prove the result

$$\max_{1\le i\le n}\|h_i - x_i\|_\infty = \mathcal{O}_{a.s.}\left\{J_s^{-p^*}(n\log n)^{2/r_0}\right\}$$

Moreover, Assumption 6 assures that

$$n^{1/2} J_s^{-p^*}(n\log n)^{2/r_0} \asymp d^{\theta/2}\left(d^\gamma C_d\right)^{-p^*}\left(d^\theta \log d^\theta\right)^{2/r_0} = d^{\theta/2 - \gamma p^* + 2\theta/r_0} O\left(\log d\right) \to 0 \quad (17)$$

Therefore, $\max_{1\le i\le n}\|h_i - x_i\|_\infty = o_{a.s.}\left(n^{-1/2}\right)$. According to the definition of $\hat{m}(\cdot)$ and $\bar{m}(\cdot)$, $\hat{m}(\cdot) - \bar{m}(\cdot)$ can be decomposed as $\hat{m}(\cdot) - \bar{m}(\cdot) = n^{-1}\sum_{i=1}^n \{h_i(\cdot) - x_i(\cdot)\}$. Lemma 2 further tells that

$$\sup_{t\in[0,1]} n^{1/2}|\hat{m}(t) - \bar{m}(t)| \le n^{1/2}\max_{1\le i\le n}\|h_i - x_i\|_\infty = o_{a.s.}\,(1).$$

Hence, $\|\hat{m} - \bar{m}\|_\infty = o_{a.s.}\left(n^{-1/2}\right)$ is proved where $\hat{m}$ is a BS estimator.

Denote by $\hat{Z}_i(\cdot) = h_i(\cdot) - \hat{m}(\cdot)$ and $\bar{Z}_i(\cdot) = x_i(\cdot) - \bar{m}(\cdot)$, we further obtain the following lemmas.

**Lemma 3.** *As* $n \to \infty$

$$\max_{1\le i\le n}\left\|\hat{Z}_i - Z_i\right\|_\infty = \mathcal{O}_{a.s.}\left\{J_s^{-p^*}(n\log n)^{2/r_0}\right\}.$$

PROOF. Denote $\hat{\phi}_k(x) = d^{-1}\mathbf{B}(x)^\top \mathbf{V}_{n,p}^{-1}\mathbf{B}^\top \phi_k$ and $\hat{Z}_i(t) = \sum_{k=1}^\infty \xi_{ik}\hat{\phi}_k(t)$ for $k \in \mathbb{N}_+$,

$$\hat{Z}_i(t) - Z_i(t) = \sum_{k=1}^\infty \xi_{ik}\left\{\hat{\phi}_k(t) - \phi_k(t)\right\}.$$

From (16) and Assumption 5, $\left\|\hat{Z}_i - Z_i\right\|_\infty \le \sum_{k=1}^\infty |\xi_{ik}|\left\|\hat{\phi}_k - \phi_k\right\|_\infty \le CW_i J_s^{-p^*}$, where $W_i = \sum_{k=1}^\infty |\xi_{ik}|\|\phi_k\|_{q,\mu}$, are i.i.d nonnegative random variables with finite absolute moment. Then

$$\mathbb{P}\left\{\max_{1\le i\le n} W_i > (n\log n)^{2/r_0}\right\} \le n\frac{\mathbb{E}W_i^{r_0}}{(n\log n)^2} = \mathbb{E}W_i^{r_0} n^{-1}(\log n)^{-2},$$

thus,

$$\sum_{n=1}^\infty \mathbb{P}\left\{\max_{1\le i\le n} W_i > (n\log n)^{2/r_0}\right\} \le \mathbb{E}W_i^{r_0}\sum_{n=1}^\infty n^{-1}(\log n)^{-2} < +\infty,$$

so Lemma 1 tells $\max_{1\le i\le n} W_i = \mathcal{O}_{a.s.}\left\{(n\log n)^{2/r_0}\right\}$. Lemma 3 is then obtained.

Next we compute the convergence rate of BS covariance estimators. For any $t, t' \in [0, 1]$, we decompose $\hat{G}(t, t') - \bar{G}(t, t')$ into three parts

$$
\begin{aligned}
\hat{G}(t, t') - \bar{G}(t, t') =& n^{-1} \sum_{i=1}^{n} \hat{Z}_i(t) \hat{Z}_i(t') - n^{-1} \sum_{i=1}^{n} \bar{Z}_i(t) \bar{Z}_i(t') \\
=& n^{-1} \sum_{i=1}^{n} \left( \hat{Z}_i(t) - \bar{Z}_i(t) \right) \left( \hat{Z}_i(t') - \bar{Z}_i(t') \right) + n^{-1} \sum_{i=1}^{n} \bar{Z}_i(t') \left( \hat{Z}_i(t) - \bar{Z}_i(t) \right) \\
& + n^{-1} \sum_{i=1}^{n} \bar{Z}_i(t) \left( \hat{Z}_i(t') - \bar{Z}_i(t') \right)
\end{aligned}
\tag{18}
$$

According to decomposition of $\{h_i\}_{i=1}^{n}$ and $\{x_i\}_{i=1}^{n}$, that is $h_i(t) = \hat{Z}_i(t) + \hat{m}(t)$ and $x_i(t) = Z_i(t) + m(t)$, we get $h_i(t) - x_i(t) = \hat{Z}_i(t) - Z_i(t) + \hat{m}(t) - m(t)$. Noting that $\hat{m}(t) = n^{-1} \sum_{i'=1}^{n} h_{i'}(t)$ and $\bar{m}(t) = n^{-1} \sum_{i'=1}^{n} x_{i'}(t)$, then $\hat{Z}_i(t) - \bar{Z}_i(t)$ can be represented by

$$
\begin{aligned}
\hat{Z}_i(t) - \bar{Z}_i(t) &= h_i(t) - \hat{m}(t) - \left\{ (x_i(t) - m(t)) - n^{-1} \sum_{i=1}^{n} (x_i(t) - m(t)) \right\} \\
&= h_i(t) - n^{-1} \sum_{i'=1}^{n} h_{i'}(t) - \left\{ x_i(t) - n^{-1} \sum_{i'=1}^{n} x_{i'}(t) \right\} \\
&= h_i(t) - x_i(t) - n^{-1} \sum_{i'=1}^{n} \left\{ h_{i'}(t) - x_{i'}(t) \right\}.
\end{aligned}
$$

Therefore, we obtain that $\hat{Z}_i(t) - \bar{Z}_i(t) = \hat{Z}_i(t) - Z_i(t) - n^{-1} \sum_{i'=1}^{n} \left\{ \hat{Z}_{i'}(t) - Z_{i'}(t) \right\}$. Hence,

$$
\begin{aligned}
& n^{-1} \sum_{i=1}^{n} \left( \hat{Z}_i(t) - \bar{Z}_i(t) \right) \left( \hat{Z}_i(t') - \bar{Z}_i(t') \right) \\
=& n^{-1} \sum_{i=1}^{n} \left( \hat{Z}_i(t) - Z_i(t) \right) \left( \hat{Z}_i(t') - Z_i(t') \right) - n^{-1} \sum_{i=1}^{n} \left( \hat{Z}_i(t) - Z_i(t) \right) n^{-1} \sum_{i=1}^{n} \left( \hat{Z}_i(t') - Z_i(t') \right).
\end{aligned}
$$

According to Lemma 3, it is easy to obtain that

$$
n^{-1} \sum_{i=1}^{n} \left( \hat{Z}_i(t) - Z_i(t) \right) \leq \max_{1 \leq i \leq n} \left\| \hat{Z}_i - Z_i \right\|_{\infty} = \mathcal{O}_{\text{a.s.}} \left\{ J_s^{-p^*} (n \log n)^{2/r_0} \right\} = o_{a.s.} \left( n^{-1/2} \right)
$$

where the last equation holds for (17). And

$$
\begin{aligned}
& n^{-1} \sum_{i=1}^{n} \left( \hat{Z}_i(t) - Z_i(t) \right) \left( \hat{Z}_i(t') - Z_i(t') \right) \\
\leq& \left( \max_{1 \leq i \leq n} \left\| \hat{Z}_i - Z_i \right\|_{\infty} \right)^2 = \mathcal{O}_{\text{a.s.}} \left\{ J_s^{-2p^*} (n \log n)^{4/r_0} \right\} = o_{a.s.} \left( n^{-1/2} \right).
\end{aligned}
$$

where the last equation follows from Assumption 6, as $d \to \infty$,

$$
n^{1/2} J_s^{-2p^*} (n \log n)^{4/r_0} \asymp d^{\theta/2} (d^\gamma C_d)^{-2p^*} (d^\theta \log d^\theta)^{4/r_0} = d^{\theta/2 - 2\gamma p^* + 4\theta/r_0} O(\log d) \to 0
$$

Hence,

$$
\sup_{t, t' \in [0, 1]} \left| n^{-1} \sum_{i=1}^{n} \left( \hat{Z}_i(t) - \bar{Z}_i(t) \right) \left( \hat{Z}_i(t') - \bar{Z}_i(t') \right) \right| = o_{a.s.} \left( n^{-1/2} \right).
\tag{19}
$$

Moreover,

$$n^{-1} \sum_{i=1}^{n} \bar{Z}_i(t') \left( \hat{Z}_i(t) - \bar{Z}_i(t) \right)$$

$$= n^{-1} \sum_{i=1}^{n} \left\{ Z_i(t') - n^{-1} \sum_{i'=1}^{n} Z_{i'}(t') \right\} \left\{ \hat{Z}_i(t) - Z_i(t) - n^{-1} \sum_{i'=1}^{n} \left( \hat{Z}_{i'}(t) - Z_{i'}(t) \right) \right\}$$

$$= n^{-1} \sum_{i=1}^{n} Z_i(t') \left( \hat{Z}_i(t) - Z_i(t) \right) - n^{-2} \sum_{i=1}^{n} Z_i(t') \sum_{i'=1}^{n} \left( \hat{Z}_{i'}(t) - Z_{i'}(t) \right). \tag{20}$$

Noting that $\left\| n^{-1} \sum_{i=1}^{n} Z_i \right\|_\infty = \mathcal{O}_p(1)$ since $\mathbb{E} \left| n^{-1} \sum_{i=1}^{n} Z_i(t) \right| \le \sum_{k=1}^{\infty} \|\phi_k\|_\infty \mathbb{E} \left| \bar{\xi}_{\cdot k} \right| < \infty$ where $\bar{\xi}_{\cdot k} = \frac{1}{n} \sum_{i=1}^{n} \xi_{ik}$. Then we have

$$\left| n^{-2} \sum_{i=1}^{n} Z_i(t') \sum_{i'=1}^{n} \left( \hat{Z}_{i'}(t) - Z_{i'}(t) \right) \right| \le \max_{1 \le i' \le n} \left\| \hat{Z}_{i'} - Z_{i'} \right\|_\infty \left\| n^{-1} \sum_{i=1}^{n} Z_i \right\|_\infty = \mathcal{O}_p \left( n^{-1/2} \right)$$

$$\left| n^{-1} \sum_{i=1}^{n} Z_i(t') \left( \hat{Z}_i(t) - Z_i(t) \right) \right| \le \max_{1 \le i \le n} \left\| \hat{Z}_i - Z_i \right\|_\infty \left\| n^{-1} \sum_{i=1}^{n} Z_i \right\|_\infty = \mathcal{O}_p \left( n^{-1/2} \right) \tag{21}$$

Substituting (21) into (20), we conclude that

$$\sup_{t,t' \in [0,1]} \left| n^{-1} \sum_{i=1}^{n} \bar{Z}_i(t') \left( \hat{Z}_i(t) - \bar{Z}_i(t) \right) \right| = \mathcal{O}_p \left( n^{-1/2} \right). \tag{22}$$

Similarly, we have

$$\sup_{t,t' \in [0,1]} \left| n^{-1} \sum_{i=1}^{n} \bar{Z}_i(t) \left( \hat{Z}_i(t') - \bar{Z}_i(t') \right) \right| = \mathcal{O}_p \left( n^{-1/2} \right). \tag{23}$$

Substituting (19), (22) and (23) into (18), we have $\sup_{t,t' \in [0,1]} \left| \hat{G}(t,t') - \bar{G}(t,t') \right| = \mathcal{O}_p \left( n^{-1/2} \right)$. Then $\|\hat{G} - \bar{G}\|_\infty = \mathcal{O}_p \left( n^{-1/2} \right)$ is proved where $\hat{G}$ is BS estimator.

PROOF OF (II) Next we prove the conclusion of BS-SPATIAL mean and covariance estimators. The estimation error of spatial mean can be computed as

$$\|\hat{m}^{\text{SPAT}} - \bar{m}\|_\infty = \max_{1 \le j \le d} \left\| \frac{1}{n} \frac{\bar{\beta}}{T(M_j)} \sum_{i=1}^{n} h_{ij} - \frac{1}{n} \sum_{i=1}^{n} x_{ij} \right\|$$

$$= \frac{1}{n} \max_{1 \le j \le d} \left\| \frac{\bar{\beta}}{T(M_j)} \sum_{i=1}^{n} h_{ij} - \frac{\bar{\beta}}{T(M_j)} \sum_{i=1}^{n} x_{ij} + \frac{\bar{\beta}}{T(M_j)} \sum_{i=1}^{n} x_{ij} - \sum_{i=1}^{n} x_{ij} \right\|$$

$$\le \frac{\bar{\beta}}{T(M_j)} \|\hat{m} - \bar{m}\|_\infty + \left( \frac{\bar{\beta}}{T(M_j)} - 1 \right) \|\bar{m}\|_\infty$$

$$= \mathcal{O}_p \left( n^{-1/2} \right)$$

where the last equality holds by noticing that $\frac{\bar{\beta}}{T(M_j)} \to_p 1$ from the law of large numbers and from Theorem 4 (i) $\|\hat{m} - \bar{m}\|_\infty = \mathcal{O}_{a.s.} \left( n^{-1/2} \right)$, $\hat{m}$ is BS mean estimator.

The estimation error of spatial covariance can be computed as

$$\|\hat{G}^{\text{SPAT}} - \bar{G}\|_\infty = \max_{1 \le j, j' \le d} \left\| \hat{G}^{\text{SPAT}}_{jj'} - \bar{G}_{jj'} \right\|$$

$$= \frac{1}{n} \max_{1 \le j, j' \le d} \left\| \frac{\bar{\beta}^2}{T(M_j) T(M_{j'})} \sum_{i=1}^{n} (h_{ij} - \bar{h}_j)(h_{ij'} - \bar{h}_{j'}) - \sum_{i=1}^{n} (x_{ij} - \bar{m}_j)(x_{ij'} - \bar{m}_{j'}) \right\|$$

$$\le \frac{1}{n} \max_{1 \le j, j' \le d} \left\| \frac{\bar{\beta}^2}{T(M_j) T(M_{j'})} \sum_{i=1}^{n} (h_{ij} - \bar{h}_j)(h_{ij'} - \bar{h}_{j'}) - \frac{\bar{\beta}_j \bar{\beta}_{j'}}{T(M_j) T(M_{j'})} \sum_{i=1}^{n} (x_{ij} - \bar{m}_j)(x_{ij'} - \bar{m}_{j'}) \right\|$$

$$+ \frac{1}{n} \max_{1 \le j, j' \le d} \left\| \frac{\bar{\beta}^2}{T(M_j) T(M_{j'})} \sum_{i=1}^{n} (x_{ij} - \bar{m}_j)(x_{ij'} - \bar{m}_{j'}) - \sum_{i=1}^{n} (x_{ij} - \bar{m}_j)(x_{ij'} - \bar{m}_{j'}) \right\|$$

$$\le \max_{1 \le j, j' \le d} \frac{\bar{\beta}^2}{T(M_j) T(M_{j'})} \left\| \hat{G}_{jj'} - \bar{G}_{jj'} \right\| + \max_{1 \le j, j' \le d} \left( \frac{\bar{\beta}^2}{T(M_j) T(M_{j'})} - 1 \right) \|\bar{G}_{jj'}\|$$

$$\le \mathcal{O}_p(1) \left\| \hat{G} - \bar{G} \right\|_\infty + o_p\left(n^{-1/2}\right) \|\bar{G}\|_\infty$$

$$= o_p\left(n^{-1/2}\right)$$

where the last equality holds by noticing that $\frac{\bar{\beta}^2}{T(M_j) T(M_{j'})} \to_p 1$ from the law of large numbers and from Theorem 4 (i) $\left\| \hat{G} - \bar{G} \right\|_\infty = o_p\left(n^{-1/2}\right)$, $\hat{G}$ is BS covariance estimator.

## G    PROOF OF THEOREM 5

PROOF OF (I) The proposed RK and RK-SPAT covariance estimators follows $\left\| \hat{G} - \bar{G} \right\| = \mathcal{O}_p(n^{-1/2})$ and the proposed BS and BS-SPAT covariance estimators follows $\left\| \hat{G} - \bar{G} \right\|_\infty = o_p(n^{-1/2})$. Consequently, the result $\left\| \hat{G} - \bar{G} \right\| = \mathcal{O}_p(n^{-1/2})$ holds for all four estimators. Moreover, it is easy to obtain that $\left\| \bar{G} - G \right\| = \mathcal{O}_p(n^{-1/2})$ where $G$ is the true covariance estimator which is usually unknown in application. In sum,

$$\left\| \hat{G} - G \right\| \le \left\| \hat{G} - \bar{G} \right\| + \left\| \bar{G} - G \right\| = \mathcal{O}_p(n^{-1/2}).$$

It is worth noticing that although although the BS and BS-SPAT estimators enjoys smaller convergence rate, $\left\| \hat{G} - G \right\|$ still converge at the rate of $\mathcal{O}_p(n^{-1/2})$ since $\left\| \bar{G} - G \right\|$ is the dominant term.

Denote $\Delta\psi_k(t) = \int (\hat{G} - G)(t, t') \psi_k(t') dt'$. Based on the result that $\left\| \hat{G} - G \right\| = \mathcal{O}_p\left(n^{-1/2}\right)$, we have $\|\Delta\psi_k\| = \mathcal{O}_p\left(n^{-1/2}\right)$ for any $k \ge 1$. Let

$$\|\Delta\| = \left\{ \iint \left( \hat{G}(t, t') - G(t, t') \right)^2 dt dt' \right\}^{1/2} = \mathcal{O}_p\left(n^{-1/2}\right),$$

then according to Hall et al. (2006), we have

$$\hat{\psi}_k - \psi_k = \sum_{j: j \ne k} (\lambda_k - \lambda_j)^{-1} \langle \Delta\psi_k, \psi_j \rangle \psi_j + \mathcal{O}\left(\|\Delta\|_2^2\right).$$

It follows from Bessel's inequality that

$$\left\| \hat{\psi}_k - \psi_k \right\| \le C\left(\|\Delta\psi_k\| + \mathcal{O}\left(\|\Delta\|_2^2\right)\right) = \mathcal{O}_p\left(n^{-1/2}\right).$$

Hence, we obtain that $\left\| \hat{\psi}_k - \psi_k \right\| = \mathcal{O}_p\left(n^{-1/2}\right)$.

PROOF OF (II) By (2.9) in Hall et al. (2006) and $\|\hat{G} - G\| = \mathcal{O}_p\left(n^{-1/2}\right)$, we obtain that

$$\hat{\lambda}_k - \lambda_k = \iint \left( \hat{G} - G \right)(t, t') \psi_k(t) \psi_k(t') dt dt' + \mathcal{O}\left(\|\Delta\psi_k\|_2^2\right) = \mathcal{O}_p\left(n^{-1/2}\right).$$

Hence, $\left|\hat{\lambda}_k - \lambda_k\right| = \mathcal{O}_p\left(n^{-1/2}\right)$ is proved.

PROOF OF (III) According to $\int \{x_i(t) - m(t)\} \phi_k(t)\, dx = \lambda_k \xi_{ik}$, we have

$$\xi_{ik} = \lambda_k^{-1/2} \int \{x_i(t) - m(t)\} \psi_k(t)\, dx.$$

Similarly, $\hat{\xi}_{ik} = \hat{\lambda}_k^{-1/2} \int \{h_i(t) - \hat{m}(t)\} \hat{\psi}_k(t)\, dt$.

For $1 \le i \le n$, $\hat{\xi}_{ik} - \xi_{ik}$ can be divided into two parts $\hat{\xi}_{ik} - \xi_{ik} = R_1 + R_2$ where

$$R_1 = \hat{\lambda}_k^{-1/2} \int \{h_i(t) - \hat{m}(t)\} \hat{\psi}_k(t)\, dt - \hat{\lambda}_k^{-1/2} \int \{x_i(t) - m(t)\} \psi_k(t)\, dt$$

$$R_2 = \hat{\lambda}_k^{-1/2} \int \{x_i(t) - m(t)\} \psi_k(t)\, dx - \lambda_k^{-1/2} \int \{x_i(t) - m(t)\} \psi_k(t)\, dt$$

We assume that for $k \in \mathbb{N}$, $\lambda_k > 0$, $\hat{\lambda}_k > 0$. Moreover, according to the fact that $\|h_i - x_i\| = \mathcal{O}_p\left(n^{-1/2}\right)$, $\left\|\hat{\psi}_k - \psi_k\right\| = \mathcal{O}_p\left(n^{-1/2}\right)$ and

$$\|\hat{m} - m\| \le \|\hat{m} - \bar{m}\| + \|\bar{m} - m\| = \mathcal{O}_p\left(n^{-1/2}\right),$$

we obtain

$$\begin{aligned} R_1 =& \hat{\lambda}_k^{-1/2} \int \{h_i(t) - \hat{m}(t)\} \left\{\hat{\psi}_k(t) - \psi_k(t)\right\} dt + \hat{\lambda}_k^{-1/2} \int \{h_i(t) - \hat{m}(t)\} \psi_k(t)\, dt \\ & - \hat{\lambda}_k^{-1/2} \int \{x_i(t) - m(t)\} \psi_k(t)\, dt \\ =& \hat{\lambda}_k^{-1/2} \int \{h_i(t) - \hat{m}(t)\} \left\{\hat{\psi}_k(t) - \psi_k(t)\right\} dt \\ & + \hat{\lambda}_k^{-1/2} \int \{h_i(t) - \hat{m}(t) - x_i(t) + m(t)\} \psi_k(t)\, dt \\ \le& \hat{\lambda}_k^{-1/2} \|x_i - m\| \left\|\hat{\psi}_k - \psi_k\right\| + \hat{\lambda}_k^{-1/2} \left(\|h_i - x_i\| + \|\hat{m} - m\|\right) \|\psi_k\| \\ =& \mathcal{O}_p\left(n^{-1/2}\right). \end{aligned}$$

Through first order Taylor expansion of $\hat{\lambda}_k$ at $\lambda_k$, it is easy to obtain that $\hat{\lambda}_k^{-1/2} = \lambda_k^{-1/2} - (1/2)\lambda_k^{-3/2}\left(\hat{\lambda}_k - \lambda_k\right) + o\left(\left|\hat{\lambda}_k - \lambda_k\right|\right)$. Hence, $\left|\hat{\lambda}_k - \lambda_k\right| = \mathcal{O}_p\left(n^{-1/2}\right)$ ensures that $\left|\hat{\lambda}_k^{-1/2} - \lambda_k^{-1/2}\right| = \mathcal{O}_p\left(n^{-1/2}\right)$. Consequently,

$$\begin{aligned} R_2 &= \left(\hat{\lambda}_k^{-1/2} - \lambda_k^{-1/2}\right) \int \{x_i(t) - m(t)\} \psi_k(t)\, dt \\ &\le \left|\hat{\lambda}_k^{-1/2} - \lambda_k^{-1/2}\right| \|x_i - m\| \|\psi_k\| = \mathcal{O}_p\left(n^{-1/2}\right). \end{aligned}$$

where $\|x_i - m\| = O_p(1)$. Therefore,

$$\max_{1 \le i \le n} \left\|\hat{\xi}_{ik} - \xi_{ik}\right\| = \max_{1 \le i \le n} \left(\|R_1\| + \|R_2\|\right) = \mathcal{O}_p\left(n^{-1/2}\right).$$

## H  MORE NUMERICAL STUDY

To visualize covariance function of functional data generated from model (4) in simulation study, Figure 4 shows the true covariance function, the sample averaged covariance estimator using original data without sparsification, and four proposed covariance estimator generated from sparsified vectors. The deviation of RK and RK-SPAT estimators is large on the diagonal $t = t'$ and the smoothness of the surfaces are poor. The accuracy of BS estimator is significantly improved, with only slight deviation at the boundary points. On the basis of RK estimator, RK-SPAT estimator further considers the spatial factor, such that the estimation accuracy of boundary points is improved but the smoothness of the surface is sacrificed to some extent.

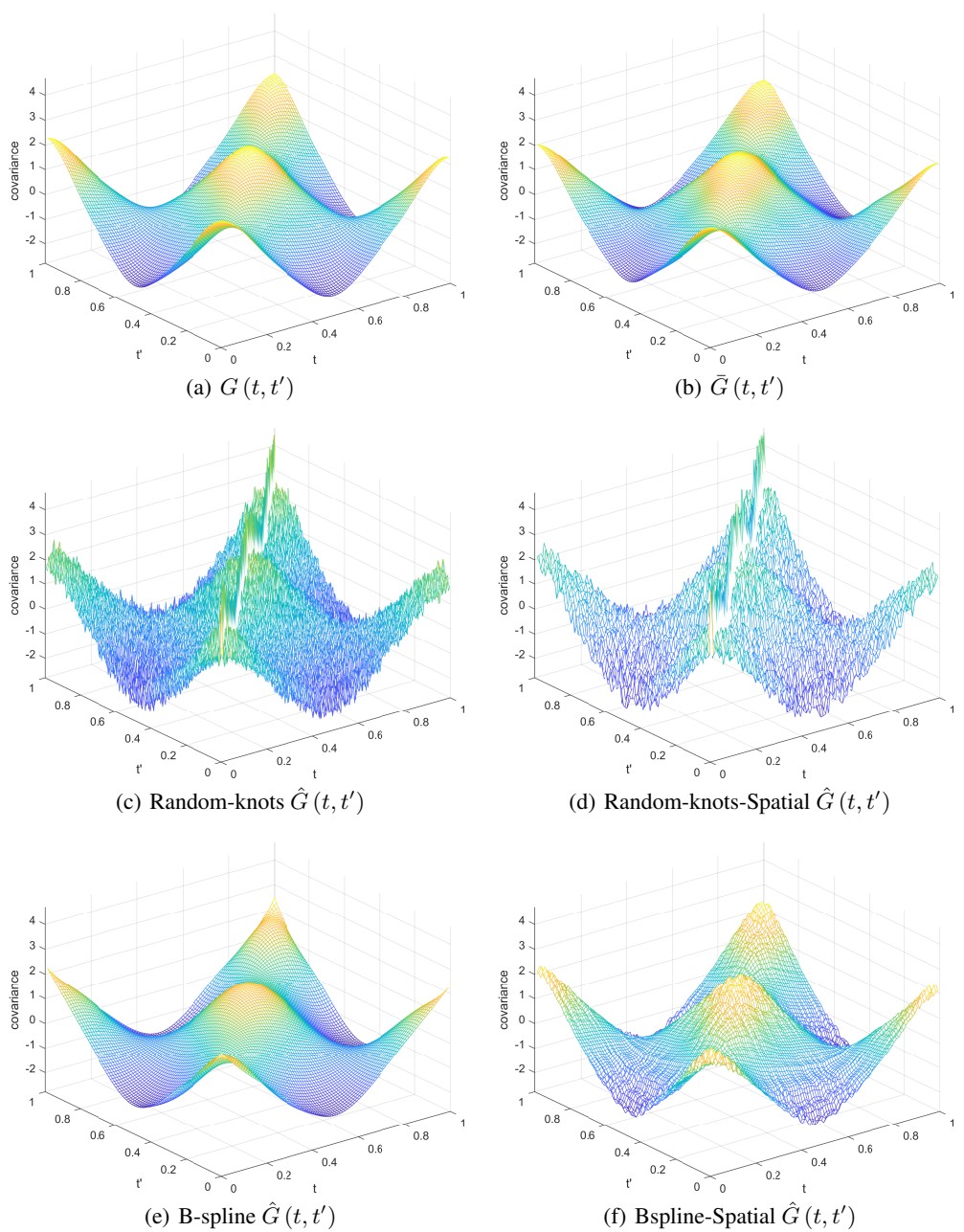

Figure 4: Plots of true covariance, averaged covariance and four different covariance estimators.

