# OpenReview forum: "Efficient Covariance Estimation for Sparsified Functional Data"
_ICLR.cc/2023/Conference — Submitted to ICLR 2023_

### Official Review · Reviewer_BUn8 · 2022-10-19

**Confidence:** 5
**Correctness:** 2
**Technical Novelty And Significance:** 3
**Empirical Novelty And Significance:** 3
**Recommendation:** 3

**Clarity, Quality, Novelty And Reproducibility:**

The main results are stated clearly. I can not say this paper is of good quality because of the problems found above. The results are new, but the six assumptions could together strongly restrict the application scenario and the novelty of the paper. The numerical computation appears to be reproducible.

**Details Of Ethics Concerns:**

The paper studies machine learning theory, and there is no ethics concern.

**Strength And Weaknesses:**

Strength:
comprehensive numerical test; theoretical results on convergence.

Weaknesses: Some typos do not belong to weaknesses, and are listed below simply because we do not find a better place to input. Below we also list some mathematical mistakes.
1. Line 5 of page 3, the notations "O" and "o" should be defined.
2. Assumption 2: for minimum value G(t,t') should be G(t,t).
3. Is "Figure 2.2" the one at the top of page 6? I feel that this figure adds confusion. Here is the reason. According to the beginning of Section 2.1, the coordinates of vectors are set to zero randomly, and for Figure 2.2(a), with probability 1-(3/6)=0.5. With such a probability and a dense vector of dim=6, we may get a sparse vector with 3 zeros, 2 zeros, 4 zeros, etc. In Figure 2.2(a), all the three dim=6 vectors are sparsified to vectors with 3 zeros. I feel uncertain about whether each element of a vector is set to zero independently, or we are just randomly select 50% of the coordinates and then set them to zero in a batch.
4. How is the norm ||G-hat - G-bar|| defined in Theorem 1?
5. At the end of page 5, it seems to me that the least square regression with B-splines would lead to interpolation. Is that true?
6. This paper proposes two families of covariance estimators, and according to Remark 2, the B-spline-based approach outperforms the random-knots family. Given such a claim, what is the reason for introducing RANDOM-KNOTS and RANDOM-KNOTS-SPATIAL?
7. Line 2 of page 12, please define the inner product <.,.>_d.
8. The first equation in Appendix B: the norm ||.|| is applied to a scalar, and should be changed to absolute value |.|.
9. The second centered equation in Appendix B could be wrong if j=j'. Please provide treatment for this case.
10. For the second centered equation in Appendix B, since we have the term \bar{h}_j which contains h_{1j}, ..., h_{nj}, and they are transformed from x_{1j}, ..., x_{nj} independently as demonstrated in Figure 2.2(a), this centered equation is wrong even if j is not equal to j'.
11. Line 4 of Appendix C, obviously xi_{ij}=1 and M_j>= 1 are not equivalent. More details are necessary to prove why the two conditional expectations are equal. The claim that event {xi_ij=1 and xi_ij'=1} happens with probability (Js/d)^2 is wrong when j=j'; this mistake may spoil the whole proof.
12. The proof of Theorem 2 is wrong because the case j=j' is not considered.
13. In the proof of Theorem 3, we note that the case p=1 is discussed. This causes 0^0 to show up in Theorem 2. Please define the value of 0^0 in such case.
14. In the proof of Theorem 3, what is the relation between (12) and (13)? In particular, if one expands the target function in (12) as a quadratic function of 1/T(r)^2, then the off-diagonal entry is in general not zero. So we believe that the proof of Theorem 3 is wrong.
15. Lines 5 & 7 on page 18: there is a factor "n" missing (but this is just a typo easy to fix). Besides this, how to derive the finiteness of the r_0 moment of W_i from Assumption 5?
16. At the bottom of page 20, as the convergence speed is not given for the second term, deriving the rate n^{-1/2} is not proper. At the end of this section there is another place with the same problem.
17. The proof of Theorem 5 fails when G has duplicated eigenvalues, in which case the estimator for an eigenvector does not converge to the eigenvector in general. Also, in line -4 on page 21, how to guarantee that C is finite?
18. On page 22, a new assumption on positive eigenvalues is added to the proof. Please also include this assumption in the theorem.

**Summary Of The Paper:**

This paper studies the problem of estimating the covariance function for a random process. The random process functional data is sampled on a finite set of knots. Four algorithms are proposed, namely, RANDOM-KNOTS, RANDOM-KNOTS-SPATIAL, B-SPLINE, and B-SPLINE-SPATIAL. These four algorithms are proposed with the main motivation of reducing computational complexity. The first two algorithms achieve this target by reducing the sampling rate, and the last two algorithms achieve this target by using B-spline interpolation. Convergence analysis of the estimated covariance function to the true covariance function is provided. Convergence of the eigensystem is also derived.

**Summary Of The Review:**

The results are new, to my best knowledge, and deserve a publication. However, some proofs are wrong, so I can not recommend a publication of this paper in its current form. I would consider to recommend a higher score after these problems have been resolved (without compromising the significance of the main results), and no more big mathematical mistake is found.

---

### Official Review · Reviewer_orUQ · 2022-10-20

**Confidence:** 4
**Correctness:** 3
**Technical Novelty And Significance:** 1
**Empirical Novelty And Significance:** 2
**Recommendation:** 5

**Clarity, Quality, Novelty And Reproducibility:**

The writing is poor. For example,
1. The mu-holder continuous functions in page 3, we cannot get the information of q from their definition.
2. Origin data set in Page 5

The problem may not be valuable. It may be better to transmit the principal components than the preprocessed data (sparsified data).

**Details Of Ethics Concerns:**

No.

**Strength And Weaknesses:**

Strength: The sparsified data costs less for transmission. The construction is simple such that the sparsified data has the similar covariance matrix get by the original data.
Weakness: The computation of covariance matrix for the sparsified data has the same (even more) complexity as that of the original data, since the mean vector (h bar) may be dense with very high probability.

**Summary Of The Paper:**

In this paper, the authors try to estimate the covariance function of sparsified functional data for dimension reduction. They propose four sparsification schemes and give the corresponding nonparametric estimation of the covariance function. The covariance estimators are asymptotically equivalent to the sample covariance computed directly from the original data. Moreover, the estimated functional principal components effectively approximate the infeasible principal components under regularity conditions.

**Summary Of The Review:**

In this paper, the authors try to estimate the covariance function of sparsified functional data for dimension reduction. They propose four sparsification schemes and give the corresponding nonparametric estimation of the covariance function. The covariance estimators are asymptotically equivalent to the sample covariance computed directly from the original data. I think it is not a good idea to transmit the original data or preprocessed data if the partner just needs the principal components. In summary, I do not think it is a good paper.

---

### Official Review · Reviewer_Dxm2 · 2022-10-25

**Confidence:** 4
**Correctness:** 3
**Technical Novelty And Significance:** 2
**Empirical Novelty And Significance:** 3
**Recommendation:** 5

**Clarity, Quality, Novelty And Reproducibility:**

The paper is well-written, and the presentation is clear. But I think the novelty is limited.

**Strength And Weaknesses:**

Strength: the paper is overall well written and the presentation is clear.
Weakness: the overall novelty of the proposed method is limited and the applicability of the B-spline-based methods may be restrictive.

1. The ideas of random sparsification and random-spatial sparsification follow rather closely to those in [1], except that [1] focuses on estimating the mean vector of the sparsified vectors. The extension to the covariance function estimation is straightforward and less innovative. If there are any nontrivial challenges in the theoretical investigations, please clarify.

2. The ideas of B-spline interpolation rely heavily on the assumption that the observed random trajectories are smooth functions. But in practice, it is commonly the case that these trajectories are contaminated with measurement errors. I am not sure whether the proposed theory can be modified to address this issue, which will be of more practical interest. I don't think this will be an issue for the random sparsifications scheme, but not sure about the B-spline interpretations.

3. I found some of the assumptions are not so standard. For example, (1) in Assumption 2, why would you need to assume $\min_{t}G(t,t')>0$, which seems to be a strange condition? (2) In Assumption 3,  it is also strange to assume the decaying rate of the eigenvalues depends on the sample size $n$.  The decaying rate is usually only assumed to change with $k$. (3) In Assumption 5, I am confused by the wording "the number of distinct distributions...is finite". Please clarify the definition.

4. I am also not so sure about some of the theoretical results regarding the convergence rate. (1) In Theorem 4, why the convergence rate for the first mean function is almost sure convergence while all other threes are convergence in probability? (2) The convergence rate in Theorem 5 appears to be faster than the existing convergence rate in the literature. For example, in Corollary 3.7 of [2], the convergence rates of these quantities are proved to be of the order $O_p(\sqrt{\log(n)/n})$, slower than those in Theorem 5. Please double-check.

Reference:

[1] Jhunjhunwala, Divyansh, Ankur Mallick, Advait Gadhikar, Swanand Kadhe, and Gauri Joshi. "Leveraging Spatial and Temporal Correlations in Sparsified Mean Estimation." Advances in Neural Information Processing Systems 34 (2021): 14280-14292.

[2] Li, Yehua, and Tailen Hsing. "Uniform convergence rates for nonparametric regression and principal component analysis in functional/longitudinal data." The Annals of Statistics 38.6 (2010): 3321-3351.

**Summary Of The Paper:**

This paper proposes two classes of sparsification schemes for densely observed functional data with potential spatial correlations. The first class of schemes is random sparsification methods that extend the work of [1] from mean estimation to covariance estimation and the second class involves B-spline interpolation of the original data. Both classes achieve the desired convergence rates as by the originally observed functional data.

Reference:

[1] Jhunjhunwala, Divyansh, Ankur Mallick, Advait Gadhikar, Swanand Kadhe, and Gauri Joshi. "Leveraging Spatial and Temporal Correlations in Sparsified Mean Estimation." Advances in Neural Information Processing Systems 34 (2021): 14280-14292.

**Summary Of The Review:**

I think the paper is well written, but in my opinion, the contributions are incremental at best.

---

### Official Review · Reviewer_DGgK · 2022-10-26

**Confidence:** 2
**Correctness:** 4
**Technical Novelty And Significance:** 3
**Empirical Novelty And Significance:** Not applicable
**Recommendation:** 6

**Clarity, Quality, Novelty And Reproducibility:**

I found this paper rather difficult to read.  Quite a lot of results are packed into the eight pages (with all proofs in the supplementary materials); but there are so many theorems that there's not so much room left for exposition!  Not even a sketch of the idea of the proof is in the main body.

On a more minor note: the labels on the figures are too tiny to be at all visible.

**Strength And Weaknesses:**

The combination of using correlation among draws and using interpolation via B-splines is an interesting theoretical idea, and the theorems look like the types I would hope for (I have not checked the details).

At the same time, despite the claimed advantages in terms of computational efficiency, this seems to mostly offer theoretical contributions.  None of the experiments addressed the claimed computational efficiency -- all were about accuracy of the estimators.

I had a hard time understanding what was happening with the application section, both in terms of the actual data to which the estimators were being combined and how that data was used.  It seems like the data was a sequence of embedding vectors; but how is it reasonable to model these as identically-distributed time series with smooth correlation?  And what happens afterward in the clustering?  It would be useful to provide a few more details.

Also, while accounting for the correlation across draws (or components or spatial locations, as the authors have it) definitely decreases the mean squared error in estimating the standard estimator using all the data, it is less clear that it decreases the mean squared error in the estimator as a whole.  The highly-correlated case is exactly the one in which the standard covariance estimates should be most noisy, I think?  It would again be useful to have a comment on this.

**Summary Of The Paper:**

The authors consider the problem of covariance estimation for a 1D time series based on $n$ identically distributed (but possibly correlated) draws of length $d$.  The standard estimator would look at all the data, but the authors consider four sparsified versions of the estimator based on a sample of the entries of the different series.  Two of the estimators use random selection of the data (one taking into account correlation between the different draws), and two use samples at fixed locations and fill in intermediate values using B-splines (again, with one taking into account covariance between the draws).  The authors also state convergence theorems for the estimated covariance and its principle eigenvalues and eigenvectors.

**Summary Of The Review:**

The paper is largely interesting for the theoretical results that it proves about the four proposed estimators.  The claimed efficiency improvements are never really demonstrated, and I found it hard to figure out what was going on in the one real application.  As a piece of mathematics, the results seem plausible and somewhat interesting, particularly in the version that incorporates both temporal and spatial correlations.  At the same time, the paper is so packed with theorems (proved in quite a long additional supplement) that it is not altogether easy to follow.

---

### Decision · Program_Chairs · 2023-01-20

**Decision:**

Reject

**Justification For Why Not Higher Score:**

There are a number of concerns not resolved as an author response is missing. The ideas in this paper may hold value but must be further developed

**Justification For Why Not Lower Score:**

N/A

**Metareview: Summary, Strengths And Weaknesses:**

This paper considers a relevant problem of estimating the covariance efficiently in the case of functional data via several sparsification schemes. There are some nice ideas that can be further developed. Reviewers were consistent in identifying several weaknesses, such as limited novelty compared to previous approaches, strong reliance on smoothness assumptions, and others detailed in the reviews. There are also a number of specific and more technical concerns that are not addressed by an author response.